


# Comprehensive space-time hydrometeorological simulations for estimating very rare floods at multiple sites in a large river basin

Daniel Viviroli[1,*], Anna E. Sikorska-Senoner[1], Guillaume Evin[2], Maria Staudinger[1], Martina Kauzlaric[3,4], Jérémy Chardon[5], Anne-Catherine Favre[5], Benoit Hingray[5], Gilles Nicolet[5], Damien Raynaud[5],
Jan Seibert[1,6], Rolf Weingartner[3,4], Calvin Whealton[7,8]

[1] Department of Geography, University of Zürich, Zürich, Switzerland
[2] Université Grenoble Alpes, INRAE, UR ETNA, Grenoble, France
[3] Mobiliar Lab for Natural Risks, University of Bern, Bern, Switzerland
[4] Oeschger Centre for Climate Change Research, University of Bern, Bern, Switzerland
[5] Université Grenoble Alpes, CNRS, IRD, Grenoble INP, IGE, Grenoble, France
[6] Department of Aquatic Sciences and Assessment, Swedish University of Agricultural Sciences, Uppsala, Sweden
[7] Paul Scherrer Institute, Villigen, Switzerland
[8] now at: Booz Allen Hamilton, Lexington Park, Maryland, United States

*Correspondence to*: Daniel Viviroli (daniel.viviroli@geo.uzh.ch)

**Abstract.** Estimates for rare to very rare floods are limited by the relatively short streamflow records available. Often, pragmatic conversion factors are used to quantify such events based on extrapolated observations, or simplifying assumptions are made about extreme precipitation and resulting flood peaks. Continuous simulation (CS) is an alternative approach that better links flood estimation with physical processes and avoids assumptions about antecedent conditions. However, long-term
CS has hardly been implemented to estimate rare floods (i.e., return periods considerably larger than 100 years) at multiple sites in a large river basin to date. Here we explore the feasibility and reliability of the CS approach for 19 sites in the Aare River basin in Switzerland (area: 17 700 km²) with exceedingly long simulations in a hydrometeorological model chain. The chain starts with a multi-site stochastic weather generator used to generate 30 realisations of hourly precipitation and temperature scenarios of 10 000 years each. These realisations were then run through a bucket-type hydrological model for 79 sub-
catchments and finally routed downstream with a simplified representation of main river channels, major lakes and relevant floodplains in a hydrologic routing system. Comprehensive evaluation over different temporal and spatial scales showed that the main features of the meteorological and hydrological observations are well represented, and that meaningful information on low-probability floods can be inferred. Although uncertainties are still considerable, the explicit consideration of important processes of flood generation and routing (snow accumulation, snowmelt, soil moisture storage, bank overflow, lake
and floodplain retention) is a substantial advantage. The approach allows to comprehensively explore possible but unobserved spatial and temporal patterns of hydrometeorological behaviour. This is of particular value in a large river basin where the complex interaction of flows from individual tributaries and lake regulations are typically not well represented in the streamflow observations. The framework is also suitable for estimating more frequent floods, as often required in engineering and hazard mapping.



## 1   Introduction

Rare to very rare floods (return periods of 1 000–100 000 years) can cause extensive human and economic damage and need to be considered in assessing flood hazard and risk to major infrastructure, as well as in safety assessments for dams. Given the immense importance of flood estimates for security and costs of hydraulic engineering measures, there is high demand for reliable information on the magnitude and shape of flood events, particularly when low probabilities are in focus. However, the comparatively short available streamflow records are a limiting factor for estimates of such low-probability floods. One common approach to derive design floods for safety assessments is to perform conventional frequency analysis on observed streamflow records, and then apply a simple return period conversion factor given by design codes (e.g., Bundesministerium für Land- und Forstwirtschaft, Umwelt und Wasserwirtschaft and Technische Universität Wien, 2009; Bundesamt für Energie, 2018; International Commission on Large Dams, 2018). In addition, it is possible to augment flood frequency analysis with additional data and evidence (Gutknecht et al., 2006; Merz and Blöschl, 2008) such as historical floods (e.g., Bayliss and Reed, 2001; Neppel et al., 2010; Hall et al., 2014; Benito et al., 2015; Salinas et al., 2016; Wetter, 2017), paleofloods (Benito and Thorndycraft, 2005; Baker, 2008; Baker et al., 2010; Benito and O'Connor, 2013; O'Connor et al., 2014), regional frequency analyses (Hosking and Wallis, 1993, 1997) or envelope curves (Castellarin et al., 2005). Also, floods can be estimated from rainfall information via simple approaches such as the GRADEX method (Guillot and Duband, 1969; Naghettini et al., 1996) or the rational method (Mulvany, 1851). Nevertheless, the comparatively short streamflow records contain a rather heterogeneous and likely unrepresentative sample of floods, and neither of the aforementioned methods is able to cover the whole gamut of possible hydrometeorological patterns and the corresponding responses of the river system. This issue has even greater relevance in large river basins, where flows from individual tributaries interact in a complex manner, possibly further complicated through flow management (e.g., lake regulation and reservoir operation).

Another common approach used in safety assessments are PMP-PMF (Possible Maximum Precipitation-Possible Maximum Flood) estimates. This approach can achieve the range of peak flow extremes examined here, but results have no clear estimate of return period and are usually not applicable over large spatial domains. Moreover, the estimation of PMP and ensuing PMF bears substantial simplifications and considerable uncertainties (Salas et al., 2014; Micovic et al., 2015; Ben Alaya et al., 2018; Zhang and Singh, 2021).

To avoid these limitations and better link flood estimation with physical processes, continuous simulation (CS) can be employed. Beven (1987) was one of the first to recognise the potential of this compelling approach, and CS has indeed been implemented in numerous studies since. However, application in industry is still challenging due to the considerable effort necessary (see overview by Lamb et al., 2016 and references therein). In CS, precipitation data are required to perform rainfall-runoff simulations and subsequently process the simulation results with conventional frequency analyses. A considerable advantage of this approach is that it is not necessary to make assumptions about antecedent conditions of a flood event (e.g., snowpack, soil moisture, storage levels of lakes and reservoirs). Although observed series of precipitation can be used as



input (Viviroli et al., 2009b), the necessary precipitation data are typically generated at arbitrary length using stochastic approaches (Wilks and Wilby, 1999) based on historical records. If necessary, hydrologic or hydraulic routing can be applied

subsequently to account for river channels and structures as well as for flood pathways in more detail (e.g., Grimaldi et al., 2013; Lamb et al., 2016; Winter et al., 2019). Moreover, it is possible to derive spatially consistent flood risk assessments by using a flood loss model (Falter et al., 2015).

To facilitate application, semi-continuous approaches that omit the complexities of rainfall generation have been proposed. SCHADEX (Paquet et al., 2013), for example, generates possible hydrological states of a catchment in a CS using daily ob-

served precipitation and temperature as input. It then combines these states with a wide range of simple synthetic precipitation events to derive flood peaks with the help of a peak-to-volume ratio, which can be estimated from a selection of observed flood hydrographs. The approach is suitable for catchments with an area of up to 10 000 km² and adapted to mountainous regions. Another example of a semi-continuous simulation approach is the SHYPRE method (Arnaud and Lavabre, 1999, 2002) and its regionalisation SHYREG (Aubert et al., 2014). This approach combines an hourly rainfall generator with

simple event-based rainfall-runoff simulations at kilometre scale and was extensively tested in France for basins with a surface area between 1 and 2 000 km² and for return periods between 2 and 1 000 years (Arnaud et al., 2017).

Long-term, fully continuous CS offers considerable advantages to estimate rare floods in a large river basin: It avoids assumptions about antecedent conditions and their spatial patterns, and also about patterns of spatial and temporal development of flood-triggering meteorological conditions. Furthermore, a considerable diversity of spatial and temporal hydrometeoro-

logical configurations can be explored, including their combination with diverse but realistic antecedent conditions. In spite of these advantages, long-term CS has hardly been implemented in this setting to date, mainly due to the difficulties involved in developing a multi-site weather generator that produces relevant results. One notable exception is the study by Hegnauer et al. (2014) for the Rhine River at Lobith (area: ~165 000 km²) and the Meuse River at Borgharen (~21 000 km²). The authors utilised a model chain with a weather generator, a catchment runoff model and routing (partly hydrologic, partly hydro-

dynamic) to provide 50 000 years of CS and subsequently derive the desired flood information, most importantly the 1 250-year design flood at Lobith and Borgharen. Following the study goals, a multi-site weather generator was implemented, and a daily time-step was used throughout. Since the generator was based on the nearest neighbour method, it could not generate precipitation amounts outside the observed range. This limitation to observed precipitation amounts was deemed acceptable since larger daily extreme precipitation had no discernible impact on the relevant winter flood frequencies (Leander and

Buishand, 2009). For sub-basins in the Swiss part of the Rhine River basin (including the Aare River basin) the authors found poorer performances and higher uncertainties than for sub-basins in other regions of the Rhine River basin, likely due to this limitation in weather generation and the use of a daily rather than an hourly time step.

What is still missing at present is a comprehensive evaluation of CS for multiple sites and at high temporal resolution in a large river basin, with focus on rare and very rare floods. Here we examine whether it is possible to use CS in this setting to

1) make reliable estimates for floods with a return period of 1 000–10 000 years, 2) derive useful information for floods with a return period of up to 100 000 years, and 3) achieve consistent estimates for more frequent floods with a return period of


10 –1 000 years. The Aare River basin, Switzerland (area: 17 700 km²) serves as a study basin. In this basin, estimates of rare to very rare floods and corresponding hydrographs are of interest at several critical sites with high (dams, weirs) or even catastrophic (nuclear power plants) damage potential, as examined in the EXAR (hazard information for extreme flood
events on the rivers Aare and Rhine) project (Andres et al., 2021).

For the present study, we coupled a multi-site stochastic weather generator, a bucket-type hydrological model and a hydrological routing system to produce 30 realisations of hourly, continuous runoff simulations with a length of 10 000 years each. In contrast to previous studies, we simultaneously attain a high temporal resolution, use exceedingly long CS, and cover numerous sites in a large river basin. This enabled us to examine the value of the hydrometeorological simulation results over a
number of temporal and spatial scale ranges, and to assess their plausibility comprehensively. In addition, we put focus on the diversity of hydrometeorological patterns represented. That being said, it has to be kept in mind that the possibilities to rigorously assess the results are limited due to the scarcity of information on rare to very rare flood events, while uncertainty analyses are hampered by the considerable computational cost of long hourly simulations at multiple sites.

## 2    Study area and observational data

With a surface area of roughly 17 700 km², the Aare River basin is one of Switzerland's major hydrological catchments and covers approximately 43% of the country. It has shares in the Alps, the Swiss Plateau and the Jura Mountains and spans an elevation range from 4 274 m a.s.l. (Bernese Alps) to ~310 m a.s.l. (confluence with Rhine River), with a mean elevation of 1 050 m a.s.l. Important land-use categories include pasture (36% of surface area), forests (30%), sub-alpine meadows (14%), bare rock (8%) and glaciers (~2%). Streamflow is heavily managed through regulation of the large pre-alpine lakes of
Biel, Brienz, Lucerne, Murten, Neuchâtel, Thun and Zurich, as well as through several dams for the production of hydroelectricity. Moreover, the river network is considerably altered from its natural state by some large corrections and a large number of smaller hydraulic structures (Schnitter, 1992; Vischer, 2003; Hügli, 2007).

For the present study, we subdivided the Aare River basin into 79 meso-scale sub-catchments with a median surface area of 123 km² (range: 19.1–1 061 km²) (Figure 1). These sub-catchments were the basis for the hydrological modelling (Section
3.3) and encompass regimes dominated to a varying degree by glaciers, snow and rain (Weingartner and Aschwanden, 1992).

The main sources of data were meteorological and hydrological records from stations operated by the Swiss Confederation (Federal Office for the Environment, 2016; MeteoSwiss, 2016) and by cantonal agencies. The meteorological data encompass continuous records of daily precipitation (1930–2014) at 105 sites, daily temperature (1930–2014) at 26 sites, hourly
precipitation (1990–2014) at 65 sites and hourly temperature (1990–2014) at 67 sites (Supplementary Table 1). In addition, an extended dataset of daily precipitation records (1864–2014) at 666 sites was available, although it contains many missing values. For the period with only daily continuous records (1930–1989), hourly precipitation and temperature values were



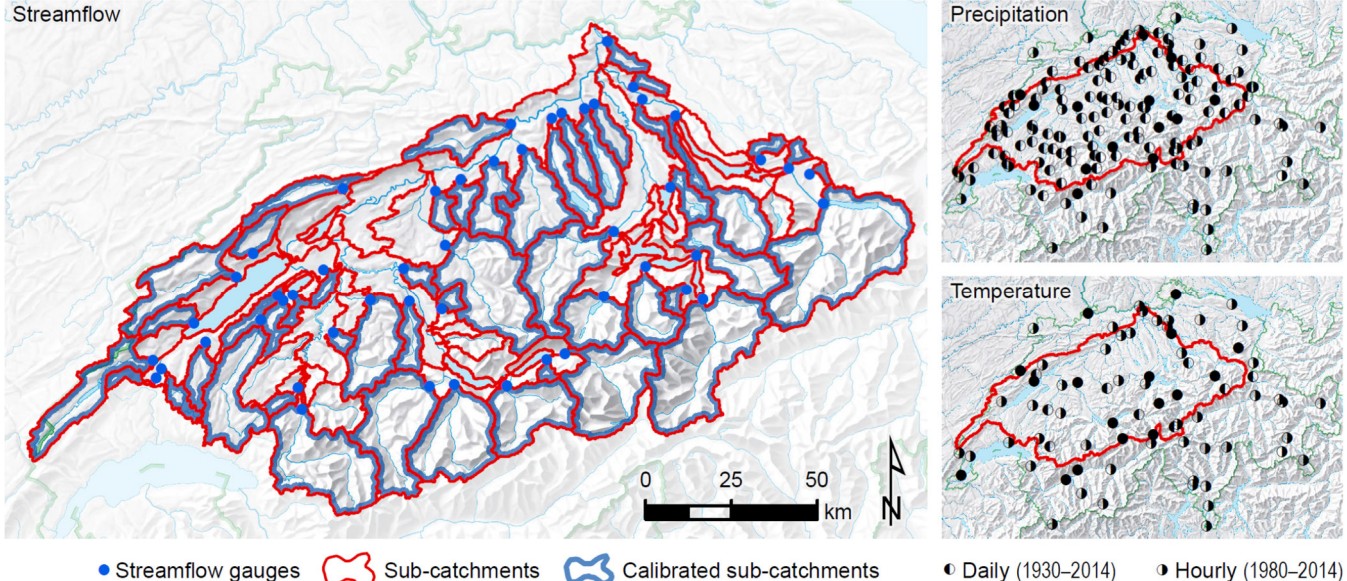

**Figure 1.** Overview map of the Aare River basin, showing data series available for streamflow (left), precipitation (top right) and temperature (bottom right). The streamflow map also reveals the 79 sub-catchments used in hydrological modelling, and which of these were calibrated.

obtained by disaggregation. This disaggregation used the temporal structure of the respective variable observed, either for the same day if available at a nearby station, or for an analogous day in the period with hourly continuous observations. The

analogy was specified using surface weather for the region, and applying constraints to preserve season and class of intensity following Breinl and Di Baldassarre (2019). The hydrological data encompass continuous discharge records at 65 stations (Supplementary Table 2). The median length of the records with hourly resolution was 36 years, with a range of 15–41 years within the period 1974–2014. The median length of the records with daily resolution was also 36 years, but with a range of 16–85 years within the period 1930–2014. In addition, records of annual maximum floods were available for some of these

stations. These records date back even further, with a median length of 94 years and a range of 32–111 years.

Several detailed hydraulic simulations with BASEMENT (Vetsch et al., 2018) were available from the EXAR project, covering relevant sites along the main branches of the Aare River system (Pfäffli et al., 2020). These simulations represent the behaviour of the river system at flows with return periods of 100, 1 000 and 10 000 years, particularly as regards bank overflow and floodplain retention (Staudinger and Viviroli, 2020).

Regulation rules for the large lakes (Lakes Brienz, Thun, Biel, Lucerne, Zug and Zurich) were provided by the corresponding authorities. Depending on the lake, the rules are aimed at diverse and partly contradicting targets such as protecting settlements downstream from floods, avoiding inundation of the lakeside areas, preserving habitats, keeping natural stage fluctuations and ensuring lake navigation. The information available ranged from detailed stage-discharge diagrams at a daily or monthly scale to rough indications of target discharge values for different intervals of lake level.





**Table 1.** Reconstructed historical (1480, 1570, 1852, 1876) and observed recent (2005, 2007) peak discharges for three sites along the Aare River (see Wetter, 2015; Baer and Schwab, 2020), as well as most extensive changes made to the river network (1714, 1878) (see Vischer, 2003).

| Year | Aare at Solothurn [a] [$m^3 s^{-1}$] | Aare at Olten [$m^3 s^{-1}$] | Aare at Brugg [b] [$m^3 s^{-1}$] |
|---|---|---|---|
| 1480 (summer) | 1 650–1 750 | - | 2 400–2 700 |
| 1570 (winter) | - | - | 2 100–2 300 |
| 1714 | Diversion of Kander River into Lake Thun | | |
| 1852 (fall) | 1 400–1 500 | 1 700–1 800 | 1 900–2 200 |
| 1876 (summer) | - | - | 900–1 100 |
| 1878 | Diversion of Aare River into Lake Biel (Jura Water Correction) | | |
| 2005 (August) | 660 | 1 035 | 1 057 |
| 2007 (August) | 719 | 1 392 | 1 387 |

[a] values for 2005 and 2007 are observations for Aare at Brügg-Aegerten routed to Solothurn with BASEMENT (see Baer and Schwab, 2020; Pfäffli et al., 2020)

[b] values for 2005 and 2007 are observations from sites Aare at Murgenthal, Wigger at Zofingen and Dünnern at Olten, combined and routed to Aare at Olten with BASEMENT (see Baer and Schwab, 2020; Pfäffli et al., 2020)

Finally, reconstructions of selected historical floods were also available. Departing from a comprehensive pilot study by Wetter (2015), four historical events were analysed in more detail within the EXAR project (Baer and Schwab, 2020). The focus was on events that could be reconstructed for more than one site, cover different seasons, and represent different states of the river corrections in Switzerland (Table 1).

## 3 Methods

### 3.1 Study set-up

Our model chain consists of three main components. First, the multi-site stochastic weather generator GWEX (Section 3.2.1) provided 30 meteorological scenarios (precipitation and temperature) with a length of 10 000 years each. Also, the methodologically independent weather generator SCAMP (Section 3.2.2) was set up and used to provide the same number of scenarios. Second, the full outputs of GWEX were used as input for the bucket-type catchment model HBV (Section 3.3.1), run at

an hourly time-step for 79 sub-catchments that cover the entire Aare River basin. Selected large events as generated by SCAMP were also run through the HBV model. Third, simulation results from the individual sub-catchments were routed downstream using RS Minerve (Section 3.3.2) for a representation of the entire Aare River system. The final simulation outputs span roughly 300 000 years at an hourly time-step and cover 19 critical sites (including the Aare River outlet) as well as the outlets of the 79 sub-catchments simulated with HBV (Figure 2, Supplementary Tables 2 and 3).

The choice of models was motivated by the specific requirements of CS, namely to cover a wide range of possible meteorological and hydrological conditions rather than the high spectrum of precipitation and streamflow only. In addition, the model chain had to be suitable for mountainous environments (i.e., consider rain-snow partitioning of precipitation, and represent snowpack as well as glaciers) and allow for a considerable number of hydrological and hydraulic complexities (i.e.,



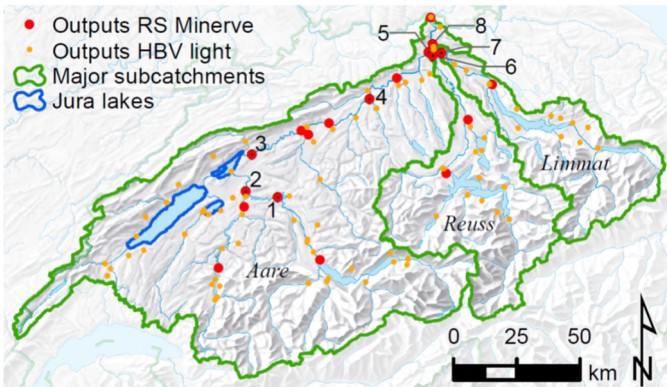

**Figure 2.** Overview map of the Aare River basin, showing model outputs of HBV light (orange) and RS Minerve (red) as well as the two major sub-basins (Reuss and Limmat Rivers) (green). Results from RS Minerve are discussed further below for the Aare River at Halen (1), Golaten (2) and Brügg-Aegerten (similar to Lake Biel outlet) (3), the Aare River at Aarburg (4) and Brugg (5), the Reuss River outlet (6), the Limmat River outlet (7) and the Aare River at Stilli (close to its outlet) (8).

lake retention and regulation, reservoir management, bank overflow, floodplain retention) to be represented. Finally yet importantly, each of the individual models had to be computationally efficient due to the vast extent of hourly simulations for a large number of sub-catchments. This requirement precluded the use of models with more detailed physical process formulations. Since all models used here have been described and tested individually, we provide only a short introduction below and refer to published literature for more details, including validation. It would certainly have been possible to use different models of similar complexity in each of the three model chain links. However, the scope of the present study was to explore

the feasibility of the CS approach for multiple sites in a large river basin at high temporal resolution, using models that have demonstrated suitability for the spatial domain and goals in focus.

### 3.2    Weather generator

### 3.2.1    GWEX

GWEX is a multisite, two-part stochastic weather generator for precipitation and temperature that relies strongly on the

structure proposed by Wilks (1998). GWEX aims to reproduce the statistical behaviour of weather events at different temporal and spatial resolutions, with a focus on extremes. Since comparatively long events are relevant in the Aare River basin, GWEX generates 3-day precipitation amounts in a first step. These amounts are then disaggregated to daily and ultimately hourly values using meteorological analogs.

The precipitation occurrence process of GWEX is represented for each site by a two-state first-order Markov chain that gen-

erates 3-day sequences with or without precipitation. The seasonality of this occurrence process is considered by estimating model parameters independently for each month of the year. Inter-site correlations between precipitation occurrence are introduced using a multivariate Gaussian distribution. For 3-day sequences with precipitation, the extended GP-Type III distribution (E-GPD) (Papastathopoulos and Tawn, 2013) is then used to generate 3-day precipitation amounts at each site. This





distribution can be described by a smooth transition between a gamma-like distribution and a heavy-tail Generalized Pareto
distribution (GPD) and has been shown to model precipitation intensities adequately (Naveau et al., 2016).

The shape parameter of the distribution is estimated with a robust regional advanced method (Evin et al., 2016) using the
extended dataset of 666 stations. Spatial and temporal dependence of 3-day precipitation amounts is represented using a first-
order multivariate autoregressive model. A Student copula represents the dependence structure of innovations in the genera-
tion process and introduces a tail dependence between at-site extremes. Similar to the occurrence process, the seasonality of
the precipitation intensity is taken into account by fitting the model for each month using a 3-month moving window.

For temperature, GWEX uses the Skew Exponential Power (SEP) distribution (Fernandez and Steel, 1998) to model the
standardized daily temperature at each station and a Multivariate Autoregressive (MAR) model to represent the spatial and
temporal dependence structures simultaneously. The seasonal cycles are accounted for with non-parametric functions and the
generation is additionally conditioned on precipitation of the current generation day.

The parameters of GWEX were defined based on daily observed weather 1930–2014, and its outputs were then further dis-
aggregated to hourly values with the help of hourly observations 1990–2014. Further details on GWEX can be found in Evin
et al. (2018, 2019).

### 3.2.2   SCAMP

SCAMP is a hybrid weather generator based on atmospheric and weather analogues and is methodologically fully independ-
ent of GWEX (Raynaud et al., 2017; Chardon et al., 2018; Raynaud et al., 2020). It generates long series of synoptic weather
over Europe in a first step, using the ERA20C atmospheric reanalysis 1900–2010 (Poli et al., 2016) as point of departure.
New atmospheric trajectories are possible by rearranging the atmospheric sequences observed within the 110 years covered
by ERA20C. A detailed description of the simulation process is given in Chardon et al. (2016) and Raynaud et al. (2020).

The resulting long series of synoptic weather are then used to generate daily weather for the Aare River basin. To this end, a
stochastic downscaling model based on atmospheric analogues is applied (Chardon et al., 2016; Raynaud et al., 2020). For
each day within the long time series of synoptic weather, the K-nearest atmospheric analogue days are identified in the ar-
chive period 1930–2010 where both ERA20C data and station records are available. The regional weather scenario for the
day in question is then generated from the statistical distribution of the regional weather observed for those K-analogues. As
shown by Mezghani and Hingray (2009) and Chardon et al. (2014), the accuracy of statistically downscaled weather scenar-
ios typically increases with spatial aggregation. In the present work, the K-nearest analogue days are thus used to generate
the regional weather, namely mean areal precipitation (MAP) and mean areal temperature (MAT) for the Aare River basin.
For the current generation day, the regional weather scenario is generated from the statistical distribution of the regional
weather observed for those K-nearest analogue days (Chardon et al., 2018). The criterion used to identify the nearest ana-
logues is a measure of the similarity of 1) the dynamic of atmospheric circulation at a synoptic scale and 2) the thermody-
namic state of the atmosphere at a regional scale. For this, we consider in a two-step identification process 1) the spatial
shape of fields of geopotential heights at 1 000 hPa and 500 hPa and 2) the mean regional-scale vertical velocity at 600 hPa





and the September–May temperature at 2 m. In summer, large-scale precipitation is used instead of vertical velocity since it has better predictive power for convective phenomena at the coarse resolution of the reanalysis data. For the hydrological simulations at sub-catchment scale, sub-daily data were needed. To this end, a non-parametric disaggregation approach was
applied, following the methodology developed by Mezghani and Hingray (2009) for the upper Rhone River and using sub-daily observations available for a limited set of stations 1990–2015.

### 3.3   Hydrological model and routing

### 3.3.1   HBV model

For the hydrological catchment runoff simulations, the HBV model (Bergström, 1972, 1992; Seibert and Bergström, 2022)
was used in the version HBV light (Seibert, 1997; Seibert and Vis, 2012). The choice of HBV was motivated by its fast processing speed (necessary for running long CS for many sub-catchments) and its well-documented suitability for flood estimation in Switzerland (Horton et al., 2021). HBV is a semi-distributed bucket-type model that uses time series of mean areal precipitation and mean air temperature as inputs. These inputs were distributed within the catchment along predefined elevation zones (here with an extent of 100 m) using constant elevation-dependent lapse rates for temperature (decrease of 0.6° C
for 100 m increase in elevation) and precipitation (a linear increase of 5% for 100 m increase in elevation, see e.g., Farinotti et al., 2012; Ménégoz et al., 2020; Ruelland, 2020). Actual evapotranspiration was estimated from the long-term daily mean of potential evaporation according to Primault (1962, 1981) in combination with observed temperature and simulated soil moisture.

The standard version of HBV consists of four main routines that represent snow processes, soil moisture, groundwater and
streamflow routing in the channel. These modules entail 15 tunable parameters. For sub-catchments with a glacier cover of 5% or more, an additional glacier routine with five tunable parameters was activated (Seibert et al., 2018). A total of 49 gauged sub-catchments (median catchment area: 115 km²; see Figure 1) were calibrated using a genetic algorithm (Seibert, 2000) with a multi-objective function that consists of Nash-Sutcliffe efficiency (Nash and Sutcliffe, 1970; weight 0.3), peak efficiency (Seibert, 2003, weight 0.5) and mean absolute relative error (weight 0.2). To account for parameter uncertainty,
100 independent model calibrations were performed for each sub-catchment and the corresponding parameter sets were retained. From this pool of 100 parameter sets, three representative sets were subsequently selected using a clustering approach to cover low, intermediate and high response in simulated peak flows as proposed by Sikorska-Senoner et al. (2020). For the remaining 30 ungauged sub-catchments, parameters were estimated from the calibrated sub-catchments using a clustering algorithm that takes the discharge regime as a discriminant and then selects two donor sub-catchments (Kauzlaric et al.,
2021). From each of these two donors, the best-performing 50 parameter sets were transferred to obtain again a total of 100 parameter sets for each sub-catchment. From these 100 sets, three representative sets were selected subsequently, as done for calibrated sub-catchments. For details on calibration, parameter set selection and regionalisation we refer to Kauzlaric et al. (2020, 2021).





### 3.3.2 RS Minerve

The complete outputs of the individual sub-catchments simulated with HBV light were finally combined to a simplified representation of the entire Aare River system using the hydrological routing system RS Minerve (García Hernández et al., 2016). As with HBV, the main reasons for using RS Minerve were its speed and well-documented applications in Switzerland (Horton et al., 2021). The routing implements major effects of bank overflow and floodplain retention (both standing and flowing) that have been parameterised based on more detailed hydraulic simulations with BASEMENT (see Section 2).

Levee breaks, by contrast, have not been implemented. Retention effects of the major lakes (Lakes Biel, Brienz, Gruyère, Lucerne, Murten, Neuchâtel, Thun, Zug and Zurich) as well as rules for their regulation (where applicable) are represented in RS Minerve, where necessary in slightly simplified form. The output nodes themselves were set at locations where river valley morphology prevents extensive floodplain inundation, and thus all discharge flows through the main river channel. This procedure was motivated by the need to partition 2D hydraulic modelling in EXAR into independent subsystems (Pfäffli et

al., 2020).

Due to the exceptionally high computational cost of long simulations at multiple sites, it was only possible to run the full set of GWEX generated weather scenarios through HBV light and RS Minerve. From the 300 000 years simulated in total, 11 000 were discarded due to technical issues, leaving 289 000 years for detailed analysis (for details, see Viviroli and Whealton, 2020). From SCAMP, 3 425 individual years containing the highest generated precipitation events were extracted

and run through the hydrological model and routing.

Note that the present implementation is not expected to be suitable for catchments with an area of less than roughly 1 000 km$^2$, both due to the initial 3-day cycle of GWEX and the hourly temporal resolution of the simulations. These specifics are unsuitable for smaller catchments where convective events become more decisive for flood behaviour (Sikorska et al., 2015).

## 4    Results

In the following presentation of results, we start by investigating the performance of each model in the model chain individually. We then proceed to results as simulated by the entire model chain, looking both at sub-catchments as well as critical sites in the Aare River system. Although results of the entire chain are decisive for assessing the reliability of the CS approach in the present context, scrutiny of the individual chain links ensures that all components of the chain produce reasona-

ble outputs and work well for the right reasons (Klemeš, 1986; Kirchner, 2006). We finally provide an overview of the spatial patterns in the most prominent events simulated for the entire Aare River basin.



## 4.1 Weather generator

### 4.1.1 GWEX

As shown by Evin et al. (2018, 2019), GWEX can reproduce the major characteristics of precipitation and temperature ob-
servations at all spatial and temporal scales considered here. Figure 3a shows the empirical return levels of maximum 1-day
mean areal precipitation (MAP1d) obtained from the 30 time series of 10 000 years each for the entire Aare River basin as
well as for five main sub-regions (see Figure 10). Figure 4a shows the same for 3-day mean areal precipitation (MAP3d). For
short return periods, for which return levels can also be estimated from observed mean areal precipitation, the return levels
of the simulations are very close to the empirical ones and highlight the good performance of the model for those variables.
For the entire Aare River basin as well as for the Neuchâtel, Thun and Aare-Emme sub-regions, most of the 18 000 year re-
turn levels were between 130 and 160 mm for MAP1d and between 190 and 225 mm for MAP3d. For the two easternmost
sub-regions (Reuss and Limmat), the values were slightly higher, between 160 and 205 mm for MAP1d and between 230
and 270 mm for MAP3d.

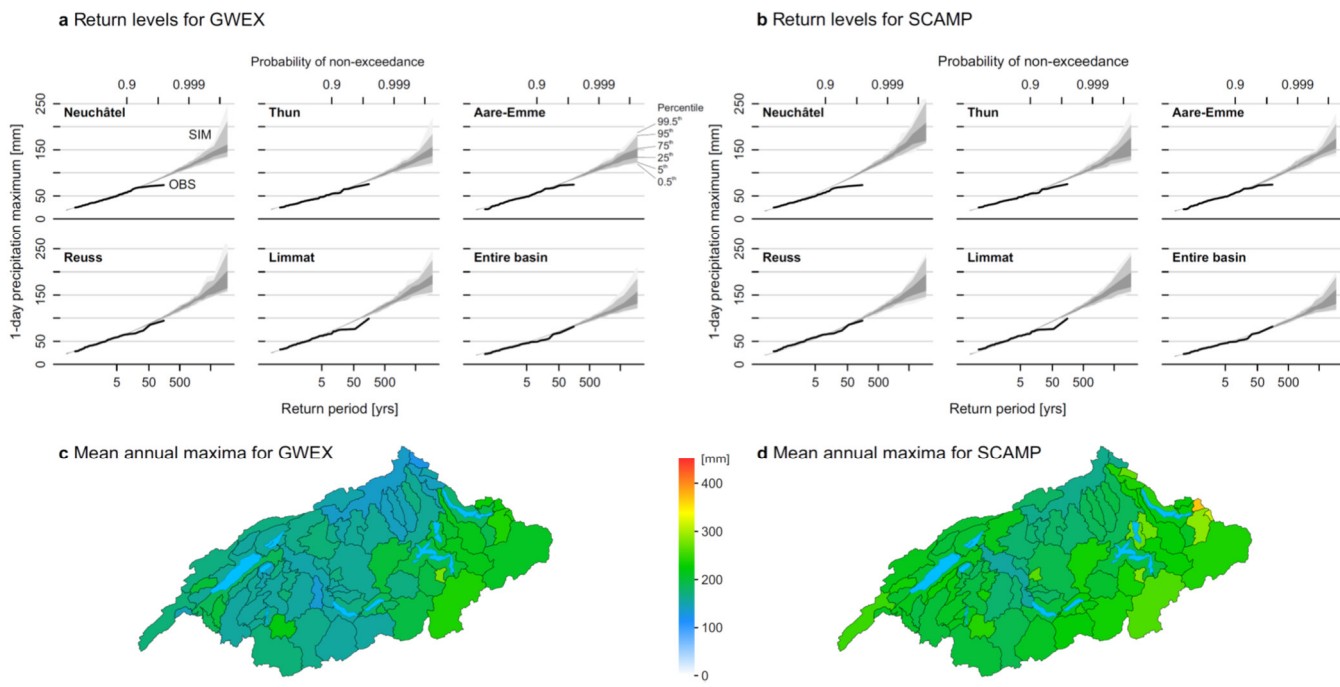

**Figure 3.** Top: Empirical return levels obtained for MAP1d (1-day mean areal precipitation) from the 30 time series of 10 000 years gen-
erated with GWEX (a) and SCAMP (b) for the entire Aare River basin and five main sub-regions using the Gringorten plotting position.
The bounds of the grey shaded areas correspond to the 0.5/99.5th, 5/95th and 25/75th percentiles of the 30 time series, respectively. Bottom:
Mean maximum simulated MAP1d from the 30 × 10 000 year time series for each of the 79 HBV sub-catchments (c: GWEX, d: SCAMP);
major lakes are drawn in cyan. Note that the largest simulated MAP value in one 10 000-year long simulation corresponds to a return pe-
riod of 18 000 years (Gringorten plotting position). Also note that the extreme MAP values mapped do not necessarily occur simultane-
ously, i.e., do not correspond to one single event.


Similar patterns are visible in the mean largest GWEX values for MAP1d (Figure 3c) and MAP3d (Figure 4c) in the sub-
basins, i.e., the average of the largest events in each of the 30 different time series. The largest values were again found in
the southeast of the Aare River basin (200–280 mm for MAP1d, 280–350 mm for MAP3d). Large values were also obtained
in the Jogne River sub-catchment in the Canton of Fribourg (220 mm for MAP1d, 297 mm for MAP3d). In the west, close to
the Jura Mountains, the values were slightly smaller (160–200 mm for MAP1d, 210–270 mm for MAP3d). Similar results
were obtained for the central part of the Aare River basin. The lowest values were found in the north (120–150 mm for
MAP1d, 160–210 mm for MAP3d).

The performance of GWEX with regard to additional characteristics was also evaluated. For instance, Figure 5 highlights the
very good performance of GWEX for the estimation of 1-day and 3-day precipitation distribution at six selected representa-
tive stations, as well as for the reproduction of wet and dry spells and for the monthly precipitation amounts at different spa-
tial scales (entire Aare River basin, large sub-regions, and the six representative stations). For a detailed evaluation of the
model we refer to Evin et al. (2018, 2019).

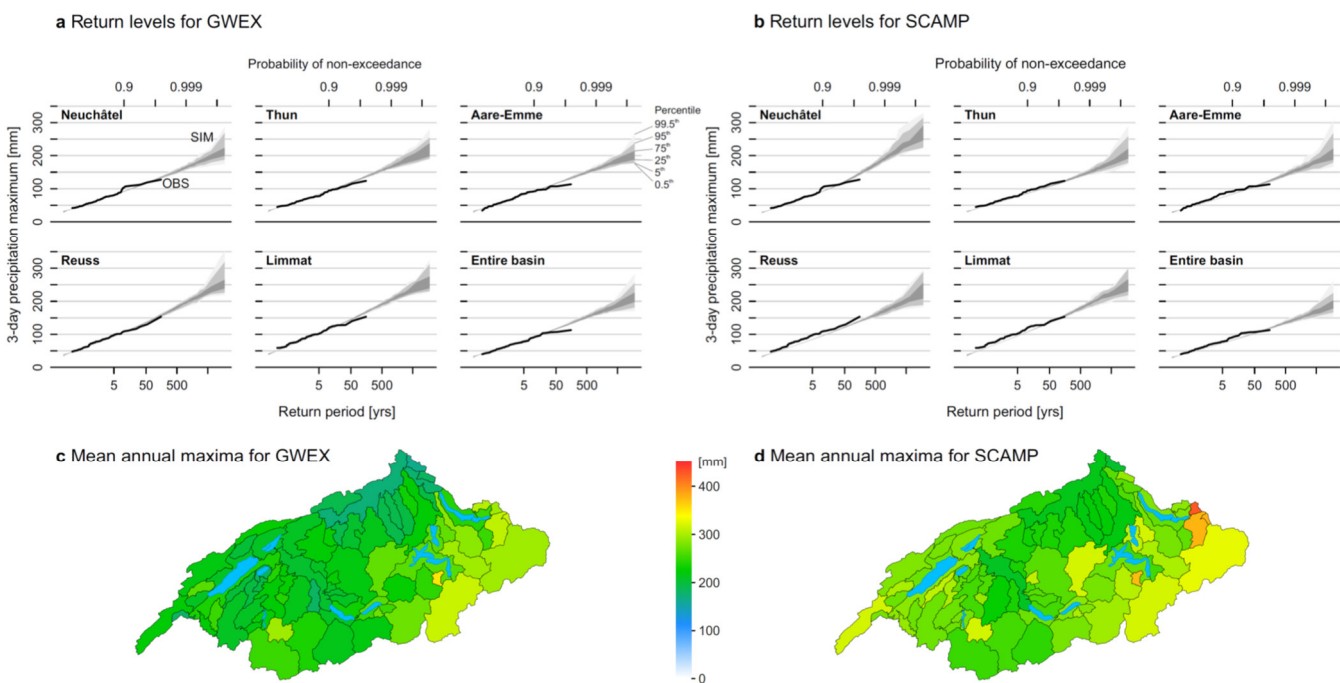

**Figure 4.** Top: Empirical return levels obtained for MAP3d (3-day mean areal precipitation) from the 30 time series of 10 000 years gen-
erated with GWEX (a) and SCAMP (b) for the entire Aare River basin and five main sub-regions using the Gringorten plotting position.
The bounds of the grey shaded areas correspond to the 0.5/99.5[th], 5/95[th] and 25/75[th] percentiles of the 30 time series, respectively. Bottom:
Mean maximum simulated MAP3d from the 30 × 10 000 year time series for each of the 79 HBV sub-catchments (c: GWEX, d: SCAMP);
major lakes are drawn in cyan. Note that the largest simulated MAP value in one 10 000-year long simulation corresponds to a return pe-
riod of 18 000 years (Gringorten plotting position). Also note that the extreme MAP values mapped do not necessarily occur simultane-
ously, i.e., do not correspond to one single event.



**Figure 5.** Multiscale evaluation of GWEX regarding monthly mean precipitation (a, b), 1-day and 3-day precipitation (c, d), dry spell lengths (e) and wet spell lengths (f). Results for panels a, c, d, e and f are for six selected representative stations, namely Andermatt (ANT), Muri (MUR), Lauterbrunnen (LTB), Courtelary (COY), Glarus (GLA) and Valeyres-sous-Rances (VAR) (see Supplementary Table 1); results for panel b are for five main sub-regions (see Figure 10) and the entire Aare River basin. Observations are drawn in black, GWEX results in grey.

### 4.1.2 SCAMP

Similar to GWEX, SCAMP can reproduce the characteristics of precipitation and temperature observations at all spatial and temporal scales considered here (see Raynaud et al., 2020 for an evaluation of SCAMP at the catchment scale and Chardon et al., 2020 for an evaluation at different spatial scales). Whatever the time scale under consideration, the 30 × 10 000-year



long weather time series also present meteorological situations that cannot be found in the observations (Raynaud et al.,
2020). For all four seasons, the ranges of simulated seasonal temperature and precipitation exceed the observed ones. For
instance, the minimum and maximum observed winter precipitation amounts are 60 mm and 490 mm, respectively. In the
SCAMP simulations, these values reached 40 mm and 690 mm. Such characteristics are particularly interesting for hydrolog-
ical purposes as they allow for simulating extreme discharge events with unobserved initial conditions in terms of soil mois-

ture and snowpack.

Results for precipitation maxima are presented on the right-hand sides of Figure 3 (MAP1d) and Figure 4 (MAP3d), for sim-
ilar spatial and temporal scales as for GWEX. Good agreement is obtained between observations and simulations for return
periods of up to 150 years, which corresponds to the maximum return period that can be estimated with the Gringorten
(1963) formula on the basis of 85 years of observed data. For the entire Aare River basin, the 18 000-year MAP1d was

140 mm on average but reached almost 200 mm for some scenarios. For MAP3d, these values reached 190 mm and 250 mm,
respectively, showing that for high precipitation events, 75% of the total amount fell within 24 hours. For both MAP1d and
MAP3d, the Limmat and the Neuchâtel sub-regions received slightly larger precipitation events, with an additional 20–
40 mm compared to the other sub-regions. This is even more visible from the return level maps associated with the maxi-
mum return periods for the 79 sub-catchments. Similar to the results of GWEX, the higher precipitation values are located on

the far southeast of the Aare River basin and the western part of the area, close to the Jura Mountains. Noticeable are the
large differences from one sub-catchment to the other, with amounts ranging from 150–350 mm for MAP1d and 200–450
mm for MAP3d. This uneven spatial structure is also visible in the observations for the 150-year return period.

At the scale of the entire Aare River basin, MAP extremes are roughly similar for GWEX and SCAMP (Figure 3, Figure 4).
At the sub-basin scale, the extremes of SCAMP are slightly larger than those of GWEX in the western part. However, the

largest difference – found for the Neuchâtel sub-region – is moderate (+10% for MAP3d and +20% for MAP1d). An im-
portant result is thus that the two weather generators produce similar large precipitation amounts at all temporal and spatial
aggregation scales considered, despite their very different modelling approaches. A further comprehensive evaluation of pre-
cipitation time series generated with both weather generators is found in Evin et al. (2018, 2019) and Chardon et al. (2020),
as well as in Raynaud et al. (2020), which reports on severity, spatial and temporal dynamics, and meteorological relevance

of events.

### 4.2   Hydrological model

Hydrological simulations for the individual HBV sub-catchments were evaluated based on three criteria: the Nash-Sutcliffe
(Nash and Sutcliffe, 1970), the Kling-Gupta (Gupta et al., 2009) and the non-parametric Kling-Gupta (Pool et al., 2018) effi-
ciencies. These criteria indicated at least acceptable results in most cases (Figure 6a) with reference to hourly discharges in

the period 1974–2015 (effective length of records per station see Supplementary Table 2), meaning that the overall stream-
flow behaviour was simulated reasonably well. The sub-catchments with poor results have widespread occurrence of karstic





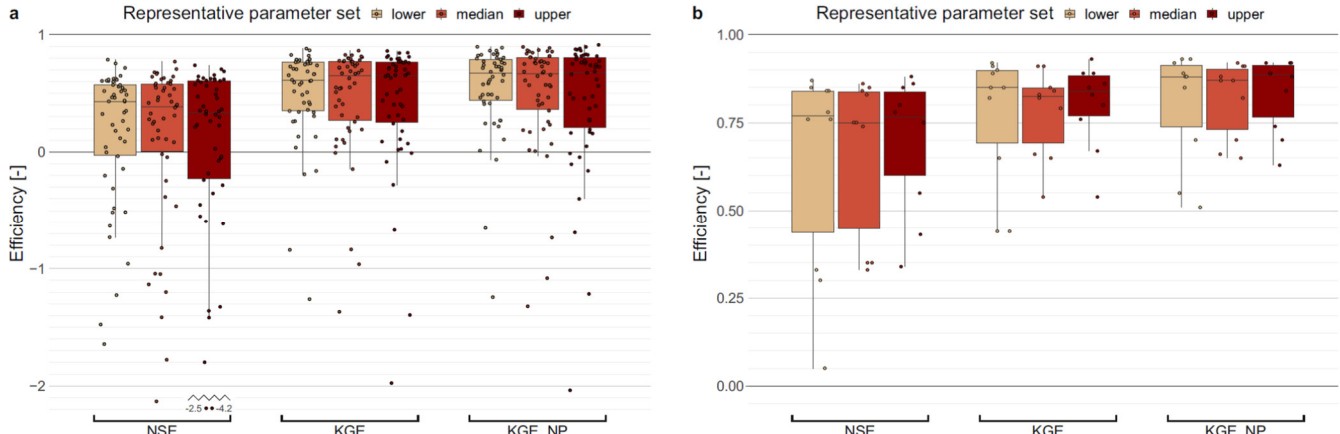

**Figure 6**. Model efficiencies at hourly time-step over the period 1974–2015 for HBV (a, 49 gauged sub-catchments) and RS Minerve (b, 10 gauged output nodes) showing the three representative parameter sets. The efficiency criteria used are Nash-Sutcliffe (NSE), Kling-Gupta (KGE) and non-parametric Kling-Gupta (KGE_NP). All criteria have an upper bound of 1 (which means ideal performance), and are unbounded towards the bottom.

rock or are affected by regulated lakes. Both of these influences are not depicted explicitly in the HBV model. The three representative parameter sets achieved rather similar median efficiencies, with the upper representative parameter set showing a tendency towards a larger spread.

An evaluation for the largest observed flood events regarding peak and volume (May 1986, June 1987, July 1987, August 1987, May 1994, May 1995, May 1999, August 2005, August 2007) showed absolute differences mainly in a range as narrow as ±1 mm h$^{-1}$. Within this range, the larger sub-catchments showed more minor deviations than the smaller sub-catchments, meaning differences are smaller in the sub-catchments that contribute high discharge to the overall Aare River basin. The largest deviation was found for the Steinenbach River catchment (range -3 mm to +2 mm), which is the smallest sub-

catchment considered in the study (surface area: 19.1 km²). The higher reliability of results for large catchments could be due to more precipitation stations being available for interpolation in space (Girons Lopez et al., 2015). Results did not show systematic patterns of some events being simulated less accurately than others. In addition, none of the three representative parameter sets showed clearly worse or better performance than any other.

### 4.3 Hydraulic routing

#### 4.3.1 Individual evaluation

The hydrological routing was first validated individually for the output nodes of RS Minerve in the Aare River system. For this, synthetic events with an estimated return period of 10 000 years were fed directly into the relevant river stretches in RS Minerve. The peak flow values of these events were determined on the basis of the regional statistical model by Asadi et al. (2018). Results were then compared with detailed hydraulic simulations in BASEMENT using the same synthetic events as

input. Such individual comparisons without use of the HBV outputs were also performed for large observed events of the last





few decades. These comparisons showed that RS Minerve is able to reproduce the discharge behaviour of both large observed as well as even larger synthetic events (Kauzlaric et al., 2020).

### 4.3.2 Joint evaluation with hydrological simulations

The efficiency of the routing was then evaluated in more detail in combination with hydrological simulations for the observed period 1974–2014. The criteria and period of this joint RS Minerve-HBV evaluation were similar to those used for evaluating the hydrological simulations individually (see Section 4.2). The evaluation was possible for 10 sites where streamflow records were available at reasonably close distance to RS Minerve output nodes (see Supplementary Table 3). The outlets of lakes were not evaluated because lake retention and regulation strongly attenuate flow dynamics, and efficiency assessments are therefore of limited value only. Results (Figure 6b) show good to very good agreement between observations and simulations (Nash-Sutcliffe efficiencies between 0.72 and 0.93) for all sites in the Aare, Reuss and Limmat Rivers. The three sites in the Emme, Lorze, and Saane Rivers showed poorer performance. A similar joint RS Minerve-HBV evaluation was done for an extended period 1930–2014 using disaggregated meteorological data as input to HBV. While the corresponding simulations were done at hourly resolution, evaluation was only possible at daily time step because streamflow observations before 1974 were available in digital form at a daily resolution only. Efficiency criteria for this longer period (not shown) were similar or even slightly higher than for the period 1974–2014 (Kauzlaric et al., 2020).

### 4.4 Entire simulation chain

### 4.4.1 Discharge characteristics

When running the full hydrometeorological model chain with weather generator scenarios instead of observed weather, there are obviously no reference observations available for evaluating streamflow results. The focus was therefore put on two selected aspects of streamflow and flood behaviour, namely cumulative frequency of streamflow and seasonality of AMFs. To assess cumulative frequency of streamflow, flow duration curves (FDCs) of simulated hourly streamflow based on GWEX were computed for all 49 gauged HBV sub-catchments as well as for the 10 RS Minerve outputs for the total Aare River system close to measurement sites. For comparison, FDCs were derived from the observations that comprise roughly 30–40 years of data, depending on the gauging station. In this comparison, the HBV simulations based on GWEX were very similar to the observations for most sub-catchments (Supplementary Figure 1). Larger differences were found for two sub-catchments only: Simme at Latterbach, where uncertainties in the discharge measurements might explain the discrepancy; and Chise at Freimettigen, where karst may be responsible. The differences between the three representative parameter sets were minimal. The RS Minerve outputs for the total Aare River system proved mostly very similar as well (Supplementary Figure 2a). Larger differences were found for the Lorze River outlet (area: 289 km²), with systematically higher simulated values for more frequent flows, and lower simulated values for high flows. Also at this site, the differences between the three representative parameter sets were largest, while they were generally small for all other sites examined. Plotting FDCs of the





highest 10% of simulated flows (Supplementary Figure 2b) revealed a general tendency to higher simulated discharge for Aare at Brügg-Aegerten and again further downstream for Aare at Brugg, stemming from rather high simulated discharges of large tributaries (Saane and Emme Rivers, respectively).

To assess flood peak characteristics, we computed the seasonality of AMFs using GWEX scenarios as an input and compared this against similar simulations using disaggregated observations as an input (see Staudinger and Viviroli, 2020). The seasonality was evaluated via the Julian date on which the annual maximum flood (AMF) occurs, and the variability of the AMF occurrences was quantified with a dimensionless measure of the spread of the data (Burn, 1997). The analyses were first done for each HBV sub-catchment independently, meaning that in each sub-catchment a different event may have been

classified as the AMF. As the difference between the three representative parameter sets mainly affected the magnitude of the AMFs but not their seasonality or the time of the occurrence, the seasonality was analysed for the median representative parameter set only. For comparison, we computed the seasonality of the simulations with disaggregated weather observations (1930–2014). Comparison (Supplementary Figure 3a) shows similar results for most of the sub-catchments. The differences that appear in some sub-catchments should not be overemphasised, since the sample size of the GWEX-based run

(AMFs from 289 000 years of simulation) was much larger than that of the run based on disaggregated observations (AMFs from 85 years of simulation), meaning that the latter contains a comparatively small subset of possible events and corresponding seasons. Overall, however, the larger picture of seasonality has a comparable pattern. The seasonality patterns for sites in the Aare River system (Supplementary Figure 3b) are strongly affected by the regulated pre-alpine lakes, and overall, a slightly earlier mean date of AMF occurrence was noted in the GWEX-based simulations with RS Minerve, except for the

outlet of Lake Lucerne (VieSee). On average, AMFs occurred 12 days earlier in GWEX-based simulations; the maximum difference was 33 days earlier. Here, it is again important to note that the sample size is different and slight disagreement should not be overemphasised.

### 4.4.2 Flood exceedance curves

The exceedance curves of AMFs derived from the full CS of 289 000 years are shown in Figure 7 for six selected sites in the

Aare River basin. For the Aare River at Halen, CS results based on GWEX are higher than observations for return periods of roughly 10 to 100 years but otherwise agree well. Downstream at Golaten, CS is generally higher than observations and extrapolations thereof. Compared to the regionally enhanced EPFL extrapolations (Asadi et al., 2018), CS also shows higher values for return periods larger than 100 years at this site, but so do the standard FOEN extrapolations of peak flow observations (Baumgartner et al., 2013). The outflow of Lake Biel then shows the strong retention effect of the Jura Lake system,

leading to a marked reduction in the largest peak discharges. This effect is visible in both observation- and simulation-based data, whereas CS is midway between the two observation-based estimates of FOEN and EPFL. Due to the further inflows downstream of Lake Biel, peak discharges increase notably again, and CS results agree very well here with the values expected from statistical extrapolation of observed events. Discrepancies occur mainly for return periods of more than 100 to




**Figure 7.** Exceedance curves for selected sites along the Aare River: Halen, Golaten (downstream of the confluence with the Saane River), outlet of Lake Biel (close to Brügg-Aegerten), Aarburg (downstream of the confluence with the Emme River), Brugg (upstream of the confluence with the Reuss and Limmat Rivers) and Stilli (downstream of the confluence with the Reuss and Limmat Rivers). Red: AMFs for CS (median representative parameter set) based on 289 000 years of GWEX weather scenarios, with the central 95% confidence interval computed according to Loucks and van Beek (2017); orange: simulated AMFs (median representative parameter set) on the basis of 85 years of disaggregated weather observations (DISAG); black: 5 highest observed peak flows plotted at return periods according to FOEN with confidence interval (Baumgartner et al., 2013) (confidence intervals that are unbounded towards high return periods are drawn in dash); blue: extrapolation of observed peak flow records according to FOEN with confidence interval (Baumgartner et al., 2013); green: regionally enhanced extrapolation of observed peak flow records according to EPFL with confidence interval (Asadi et al., 2018); light brown (for Brugg and Aarburg only): range of reconstructed historical floods (Baer and Schwab, 2020). Observation and reconstruction sites do not always match simulation sites exactly; the corresponding values have been scaled where necessary, assuming constant discharge per unit area.



1 000 years, where CS shows higher values. The simulations using disaggregated observations of temperature and precipitation for 1930–2014 show high agreement with observations as well. This is also the case for Aare at Golaten, where CS achieved higher AMFs in comparison to observations and extrapolations. A discussion of differences with explanations will
follow in Section 5.2.

### 4.4.3 Spatial patterns of largest events

For an overview of spatial variability of extremes in meteorology and hydrology, Figure 8 maps the conditions present in the ten generated events that lead to the highest peak discharges at the outlet of the Aare River basin. The data refer to GWEX scenarios (Figure 8a) and corresponding HBV simulations (Figure 8b) for the 79 sub-catchments. For the 72-hour cumula-
tive precipitation scenarios (Figure 8a), a relatively large range of conditions is found. The two largest hydrological events show a widespread occurrence of high precipitation sums with a slight emphasis on the central and eastern parts of the basin. Most of the other top-10 events have a stronger emphasis on parts of the basin where the region of emphasis varies. Simulated specific peak discharge (Figure 8b) shows stronger and more homogeneous regional accents, but still a variety of spatial patterns. Values are often somewhat lower in the southern and southeastern parts of the basin. Contributions from these
regions to peak discharges downstream are strongly attenuated by the pre-alpine Lakes Brienz, Lucerne, Thun and Zurich. Therefore, the highest peak discharges simulated downstream in the Aare River (and presented here) are not caused by large events upstream of the pre-alpine lakes, and in turn, the events with high peak discharges upstream are not well represented in the events selected here.

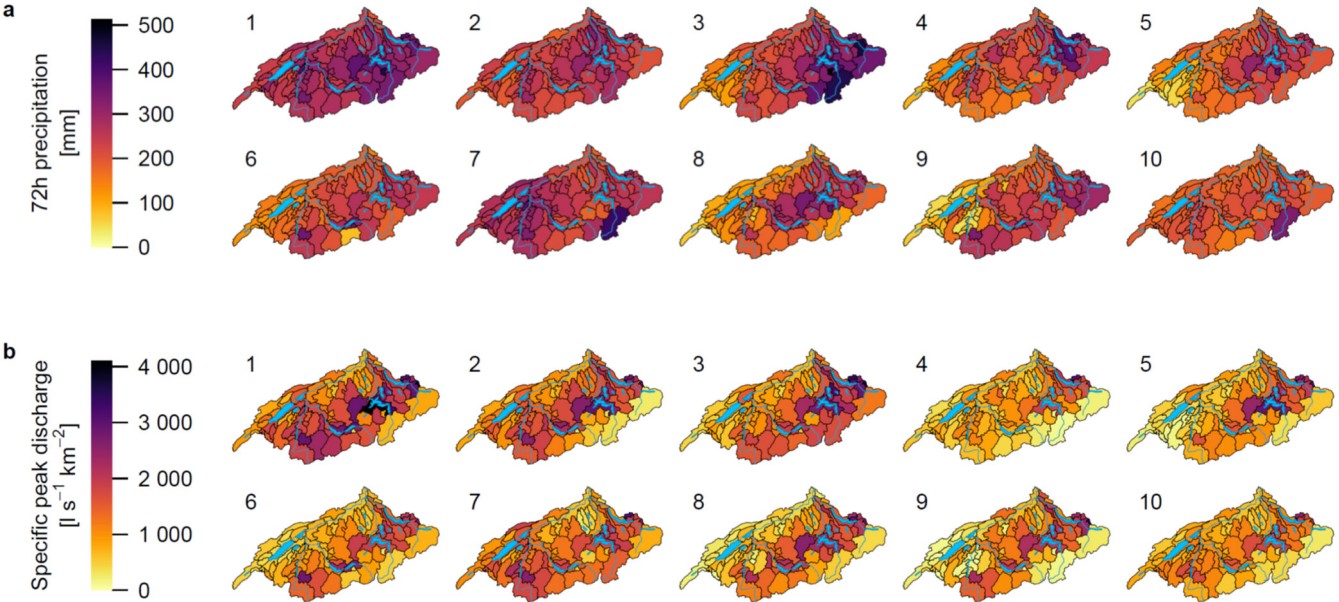

**Figure 8.** Patterns of cumulative precipitation (maximum 72-hour sum) (a) and specific peak discharge (b) for the 10 largest peak flow events simulated at the outlet of the Aare River basin. For return periods estimates see Figure 11.

Many of the simulated top-10 events have contributions of snowmelt runoff (Figure 9a), mainly originating from the alpine

area in the southern and southeastern part of the Aare River basin. In the extreme events studied here, notable snowmelt is
possible even in July or August. Weighted over all sub-catchments, the ratio of snowmelt runoff volume to total runoff vol-
ume in the 72 hours preceding peak flow at the Aare River outlet was between 7% and 31% in the ten events studied here,
with snowmelt runoff from the sub-catchments ranging between 3 and 19 mm (median: 11 mm). Alpine sub-catchments lo-
cated in the south and southeast occasionally reached ratios of more than 60%. As mentioned above, however, runoff from

these sub-catchments is strongly attenuated by the pre-alpine lakes. The ratios were much smaller in the Swiss Plateau, with
the exception of event 5. The average ratio of snowmelt runoff volume (72 hours preceding peak runoff at the outlet) to max-
imum 72-hour precipitation during the top-10 events was between 1% and 13% and showed slightly more diverse patterns
than the ratio of snowmelt to total runoff volume.

To examine antecedent conditions, the status of simulated soil moisture was assessed. Here, we considered simulated soil

moisture relative to maximum storage. It is important to note that maximum storage is a parameter of HBV and not neces-
sarily equal to measured values of field capacity. Some variability can be noted five days before peak flow at the outlet (Fig-
ure 9b), although the large precipitation amounts then led to extensive saturation in the following days. At the time of peak
flow at the Aare River outlet, nine out of ten sub-catchments had filling levels of 85% or more, and averaged over the entire
basin, the filling levels ranged between 90% and 95% for the individual events. The limited variability at the time of peak

flow at the outlet is not surprising because here, we study the largest events, which are indeed caused by a combination of

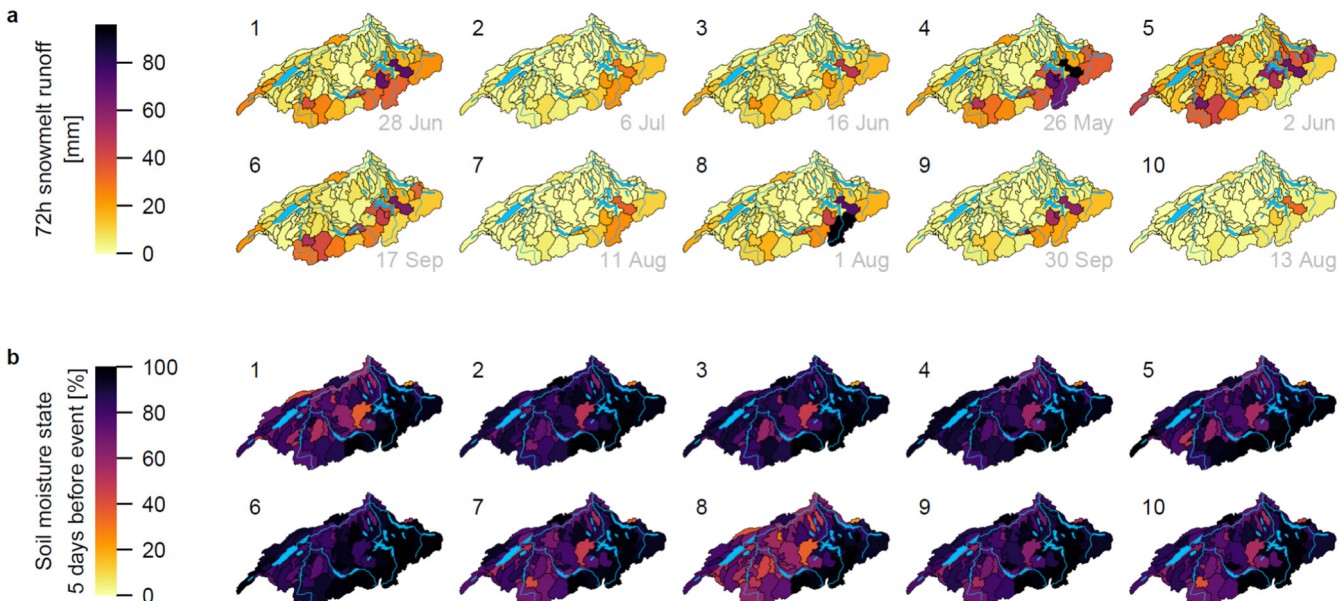

**Figure 9.** Patterns of snow melt runoff (sum over the 72 hours preceding peak flow at the Aare River basin outlet) (a) and soil moisture
storage (filling level relative to maximum storage available, five days before peak flow at the Aare River basin outlet) (b) for the top-10
peak flow events simulated at the outlet of the Aare River basin.




high soil moisture (i.e., high runoff ratio) and large precipitation amounts. By design of the CS approach, however, the broad spectrum of floods simulated also encompasses events with high saturation from considera-

ble antecedent precipitation but moderate precipitation amounts during the event, as well as events with moderate saturation but large precipitation amounts.

## 5 Discussion

### 5.1 Diversity of critical hydrometeorological configurations


A strong benefit of the multi-site, long-term hydrometeorological CS approach is the possibility to explore a vast diversity of hydrometeorological configurations and to generate critical combinations of initial hydro-

logical state patterns and weather dynamics, including combinations that can generate very rare floods. This benefit is particularly relevant when a large river basin is in focus, as in our case. Due to the large spatial extent, the number of tributaries and the complexity of hy-

draulic conditions, a wide variety of combinations is possible regarding hydrometeorological states and dynamics. However, this variety will hardly be reflected in observations since these only provide a comparatively short, arbitrary and thus most likely unrepresenta-

tive sample.

The diversity of configurations simulated by the weather generator is illustrated with the severity maps of precipitation in Figure 10. These maps present the GWEX-generated precipitation amounts of the 10 larg-

est peak flow events at the Aare River outlet. In detail, they report the return periods of cumulative precipita-

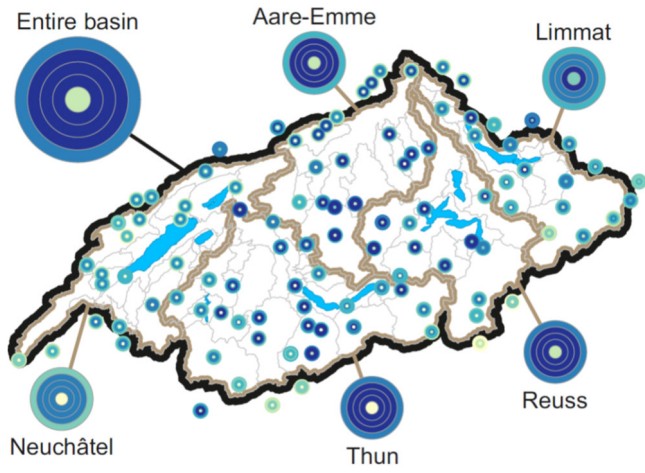

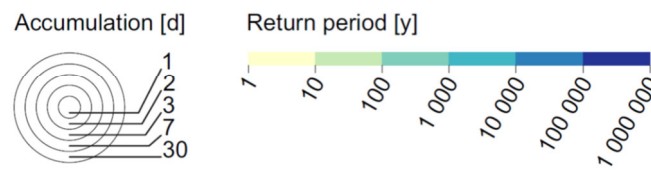

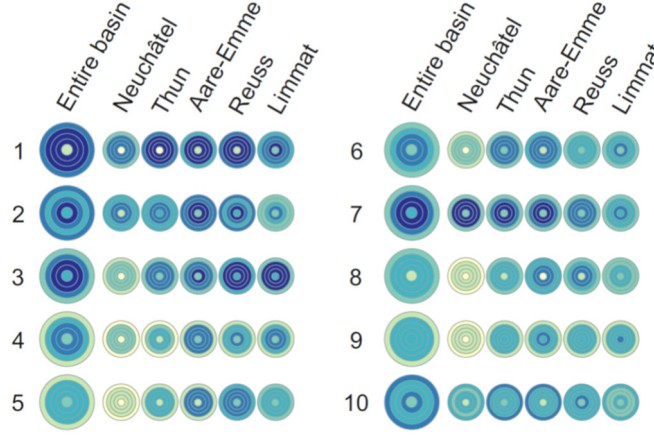

**Figure 10.** Return periods of precipitation over different accumulation durations (in days before occurrence of peak discharge at the Aare River outlet), shown for all sites considered (only for largest hydrological event in panel a) as well as for different spatial aggregations (sub-regions and entire basin) for the 10 biggest peak flow events simulated at the outlet of the Aare River basin (panel b).




tion amounts for all sites considered, as well as for different spatial and temporal scales over the 30 days preceding the flood peak at the Aare River outlet. The largest peak discharge event is clearly triggered by a very large precipitation event, the corresponding return periods exceeding 100 000 years for accumulation durations from 2 to 7 days in most of the Aare River

basin (Figure 10a). Figure 10b shows that the spatial and temporal variety of triggering precipitation is indeed very large, and the critical regions as well as the critical accumulation durations vary substantially between events. This highlights that aside from precipitation, other factors (such as the coincidence of floods from different sub-regions) are also important for the generation of the extreme floods simulated.

Further severity maps are available in Staudinger and Viviroli (2020), also covering the five largest 1-day and 3-day GWEX

precipitation events as well as all similar maps for SCAMP-generated precipitation. Analysis of these maps confirms that for both GWEX and SCAMP, a large variety of spatial and temporal dynamics were generated, consistent with the variety of events present in the observation period and beyond that exploring different combinations of antecedent precipitation and event precipitation severity. While it was not possible to check the realism of these maps quantitatively, a visual analysis together with experts from MeteoSwiss did not reveal unrealistic patterns.

For the entire model chain, spatial patterns of the top-10 hydrological were shown in Figure 8. However, this display partly masks the variety of conditions because precipitation, specific peak discharge and snowmelt runoff have inherent patterns due to climatological differences between the plateau, Jura, pre-alpine and alpine areas (see e.g., Isotta et al., 2014). These

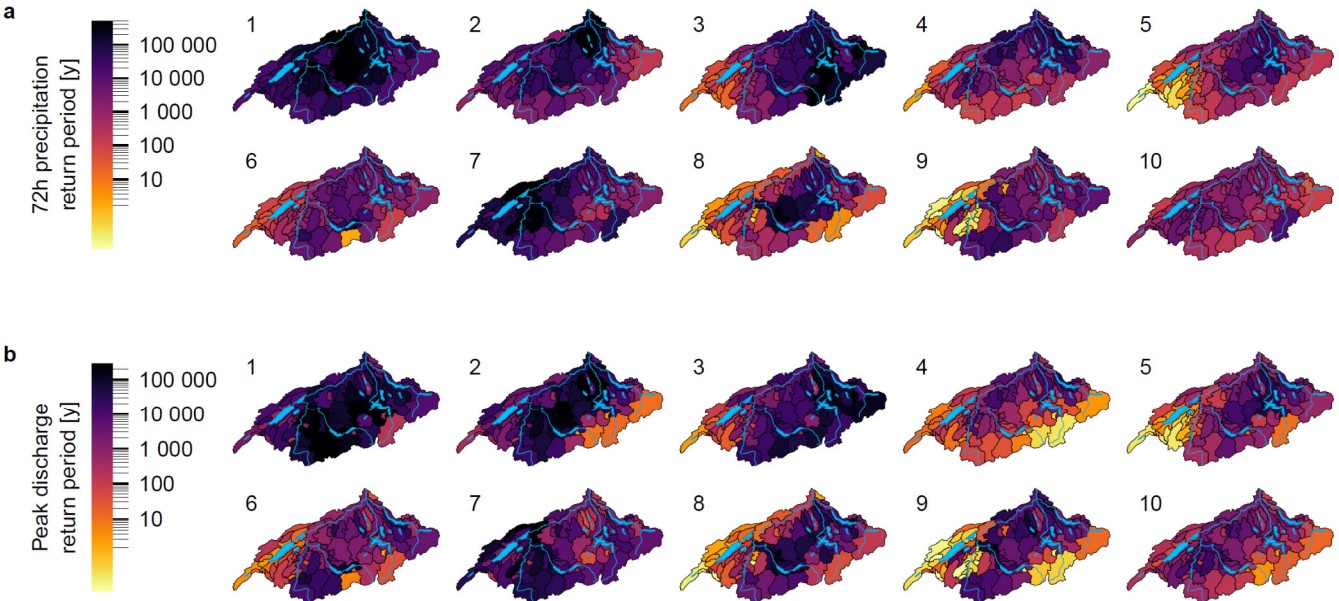

**Figure 11.** Patterns of return periods for cumulative precipitation (maximum 72-hour sum, using Gringorten plotting position) (a) and specific peak discharge (using Weibull plotting position) (b) for the top-10 peak flow events simulated at the outlet of the Aare River basin. For absolute values see Figure 8.





climatological differences can be removed by considering return periods instead of amounts, as was already done in the se-
verity maps for precipitation above (Figure 10). These return periods (Figure 11) reveal a considerable variety also for the
hydrological patterns. Note that a map for snowmelt and soil moisture return periods is not provided because the simulated
state variables were only stored for the events selected due to large disk write time and storage costs.

Overall, the largest floods in the generated time series came from very different hydrometeorological configurations: Some
of the floods were caused by huge precipitation amounts falling over the whole Aare River basin for one or two days preced-
ing the flood peak; some were caused by heavy precipitation concentrated on smaller, varying parts of the Aare River basin;
and some were caused by precipitation falling over a few days preceding the flood peak. The emerging patterns of peak
flows in the individual sub-basins are similarly diverse and essentially follow the patterns of 72h cumulative precipitation,
with some differences due to the varying spatial and temporal dynamics of the precipitation scenarios and varying contribu-
tions of snowmelt.

## 5.2 Realism of resulting floods

Although the present CS chain relies on state-of-the-art models parameterised with robust regional approaches and estima-
tion methods, its results need to be assessed and checked in some way. This concerns both the plausibility of the large flood
events obtained and the return periods estimated from the exceedingly long CS. In the following we compare the results of
the CS chain to observations and estimates obtained from previous work in the region. Although a strict comparison is not
possible for different reasons mentioned below, this analysis is nevertheless informative.

### 5.2.1 Full exceedance curve

Overall, the full exceedance curves of AMFs from the hydrometeorological model chain (Figure 7) compare well with stand-
ard statistical extrapolations of observed data (Baumgartner et al., 2013), and as well as extrapolations enhanced with a re-
gional statistical model (Asadi et al., 2018). At Halen (Figure 7a), the most upstream site considered in the Aare River, the
discrepancy for events with a return period of less than 100 years is explained by a flood discharge tunnel that was com-
pleted in 2009. While this tunnel is represented in the model, it affects only the last few years of streamflow observations
and is thus only marginally represented in the extrapolations. Roughly 17 km downstream at Golaten (Figure 7b), the simula-
tions yield higher AMFs than observations and extrapolations across all return periods, mainly because of the inflow of the
Saane River immediately upstream. In comparison to observations, the Saane River shows noticeably higher simulated
AMFs from the full CS, even though simulated AMFs from using disaggregated meteorology 1930–2014 agree well with
observations. Downstream of the Jura Lake System (Lakes Biel, Murten and Neuchâtel, see Figure 2), the simulations show
higher AMFs for return periods considerably larger than 100 years. Here, the model can simulate a failure of the Jura Lake
System, which would lead to a reactivation of the original bed of the Aare River and a bypassing of the three Jura lakes. In
such an event, the flood peak in the Aare River downstream of Lake Biel would arrive considerably faster and more pro-
nounced. This bypassing would occur at a discharge of around 1 880 $m^3 s^{-1}$, which is higher than the maximum of





1 514 $m^3$ $s^{-1}$ recorded in 2005, and is thus not represented in discharge records and extrapolations thereof. In a similar vein, the highly nonlinear response of the three lakes and their interplay with widespread inundations during extreme events is poorly sampled by the records. At Stilli (Figure 7f), downstream of the confluence of the Aare (surface area at this location: 11 708 $km^2$), Reuss (3 426 $km^2$) and Limmat (2 412 $km^2$) Rivers, the discrepancy for rare and very rare floods is likely due to unfavourable configurations of weather events that are inadequately sampled in the streamflow observations. Although the

flood peaks from the Aare, Reuss and Limmat Rivers did arrive in a relatively narrow time window of less than ten hours in some of the largest observed events (e.g., 1994, 2005 and 2007), only one of the three individual rivers showed a flood with a return period exceeding roughly 50 years in all of these events.

    As an important context for the above juxtaposition of simulated and observed values, it should be emphasised that results of the CS chain are not directly comparable to statistics of observed discharge for several reasons. First, observations of annual

maximum discharge have limited length (here between 32 and 112 years), and extrapolations are usually not recommended for return periods of more than 100–300 years due to the large uncertainties (Maniak, 2005; Baumgartner et al., 2013). Second, many streamflow records are inhomogeneous due to hydraulic structures and diversions built over the decades. One of the most important examples in the Aare River basin is the second Jura Water Correction 1962–1973 (Vischer, 2003). This correction led to slightly higher values for frequent AMFs downstream of Lake Biel. Consequently, the extrapolation for less

frequent floods gives slightly higher values as well (see Klemeš, 1986 for the impact of low values on the upper tail of a probability distribution). This inconsistency has been eliminated from the FOEN flood statistics by dismissing years before 1974 (Bundesamt für Umwelt, 2020). By contrast, it is not feasible to eliminate the impact of the many further, smaller alterations. Third, for a large river basin such as the Aare, flood configurations can derive from a multiplicity of specific hydrometeorological configurations, as already mentioned. Many of those configurations have not yet been observed. The obser-

vational period is thus much too small to provide a representative sample of possible hydrometeorological configurations. It is expected that this poor representativity is reduced with the flood sample obtained from long CS. Indeed, CS essentially exploits precipitation and temperature records (here with a length of 85 years) at multiple sites to parameterise a multi-site weather generator, and to enable exceptionally long hydrological simulations that are finally evaluated statistically. The extrapolation to small probabilities is thus based on meteorological rather than hydrological observations, and therefore a con-

siderably broader range of conditions can be covered than is present in the streamflow records. This concerns especially the extent, spatial configuration and temporal progress of triggering meteorological events and the combined hydrological response of the many sub-catchments (see previous section). All in all, there is no reason to expect a perfect statistical correspondence between observations and simulations.

    Results of CS have also been compared to selected historical flood events of the past 540 years in the region. At all sites

where it is possible to reconstruct such historical events (Table 1), the largest simulated AMFs exceed the reconstructed peaks clearly (see examples for the Aare River at Aarburg in Figure 7d and at Brugg in Figure 7e). Again, reconstructed historical and current flood peaks should only be compared with due care. On the one hand, major changes have been made to the river network over the past centuries. Under today's conditions, the peak values of historical floods would have a smaller





probability, mainly because of the diversions of the Aare River into Lake Thun (1714) and Lake Biel (1878). These diver-
sions have been made to exploit lake retention and thus attenuate flood peaks under present conditions. On the other hand,
long-term internal climate variability over time scales of decades to centuries is likely to have impacted flood frequencies
(Redmond et al., 2010). In northern Switzerland, four periods rich in floods occurred since 1500, lasting roughly between 30
and 100 years each. The current period rich in floods started in 1970, and the previous such period occurred 1820–1940
(Schmocker-Fackel and Naef, 2010a). In the CS approach used here, the low-frequency fluctuations in large-scale atmos-
pheric circulation underlying the flood-rich and flood-poor periods (Schmocker-Fackel and Naef, 2010b) have not been ac-
counted for by the weather generators, limiting in turn the comparability of return periods. Keeping in mind these limita-
tions, the realism of the highest floods simulated at all sites considered is assessed in Figure 12 in comparison to the compre-
hensive records of large observed and reconstructed flood peaks in Switzerland (Eidgenössisches Amt für Strassen- und
Flussbau, 1974; Kienzler and Scherrer, 2018). These records date back to 1342, and the earliest record for the Aare River
basin is from 1629. Maximum peak discharges from CS are higher than the recorded floods in the Aare River basin for
catchment areas of more than roughly 1 000 km², with a factor of about two for the largest catchments simulated. At some
sites, the simulations exceed the enveloping curve for maximum discharge in the Rhine River basin estimated by Vischer
(1980) on the basis of data from Eidgenössisches Amt für Strassen- und Flussbau (1974). However, they are still within the

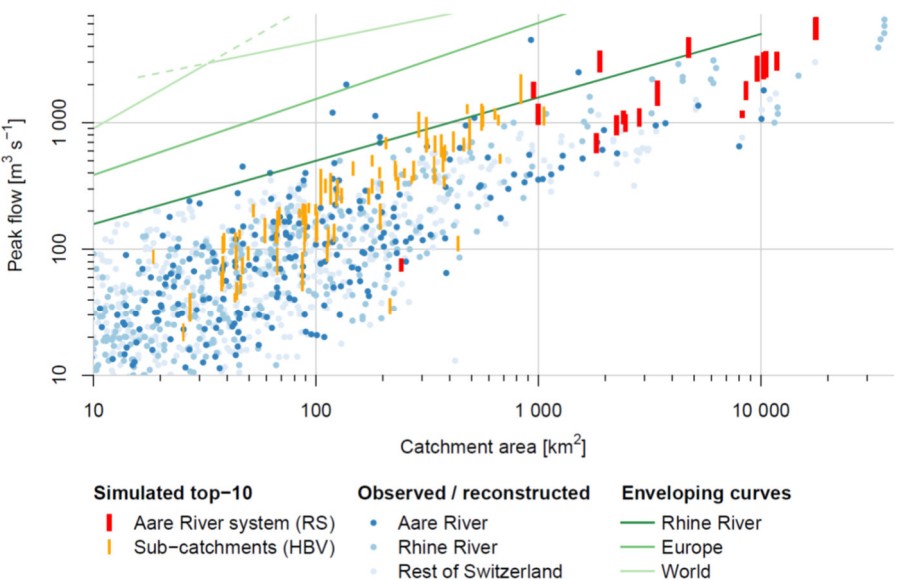

**Figure 12.** Range of the top-10 flood peaks simulated based on GWEX scenarios with the median representative parameter set for each
output node of the Aare River system (RS Minerve output, red) and each sub-catchment (HBV output, orange), in comparison to observed
and reconstructed peak discharges (Eidgenössisches Amt für Strassen- und Flussbau, 1974; Kienzler and Scherrer, 2018), separately for
the Aare River basin (dark blue), the Rhine River basin (without sites in the Aare River basin) (medium blue) and the rest of Switzerland
(light blue). For orientation, enveloping curves are shown for the Rhine River basin (Vischer, 1980, here including the Aare River basin;
valid for catchments with an area of up to roughly 10 000 km²), Europe (Marchi et al., 2010) and the world (Herschy, 2002).





range of maximum discharges recorded in other parts of Switzerland. The fact that simulated flood peaks fall noticeably be-
low the top of the point cloud for catchments with an area of less than 100 km² indicates that indeed the set-up of the weather
generator and the hourly time-step are not suitable for estimating rare flood peaks in individual smaller catchments, where
short convective events typically lead to maximum discharge. As noted in Section 3.3.2, results should only be interpreted
for catchments with an area of more than 1 000 km². At that scale, the comparatively lower values from small catchments are
not relevant, as rather the interplay of these catchments in reaction to precipitation events lasting a few days becomes deci-
sive. All values simulated lie well below the enveloping curves for Europe (Marchi et al., 2010) and the world (Herschy,
2002). However, these comparisons have limited validity due to large differences in climatological and hydrological condi-
tions.

### 5.2.2 More frequent floods

One of the main advantages of long-term CS with a hydrometeorological model chain is that it not only provides information
about peak flows with small probability, but can also lead to consistent results for more frequent floods as is often required
in engineering and spatial planning. To examine the validity of our results in this respect, we subdivided the full CS into
blocks that, in length, correspond to the length of the observed peak flow record at the site examined (e.g., Aare at Stilli:
2 580 blocks with length of 112 years) (Figure 13). Comparison with observed floods shows high agreement for the Aare
River at Aarburg (Figure 13a) and the Limmat River outlet (Figure 13c). At Aare-Aarburg, between 4 and 7% of the 100-
year simulation blocks exceeded the range reconstructed for the historical flood of 1852. For the Reuss River outlet (Figure
13b) and the Aare River at Stilli (Figure 13d), the bulk of the simulation blocks reach higher floods than the observations.
However, the observed peaks still fall within the confidence intervals of the simulation blocks. The simulations using dis-
aggregated precipitation and temperature 1930–2014 agree well with observations and CS.

In this context, it is important to remember that only the recorded peak flow is an observation (albeit subject to measurement
errors and uncertainty of the stage-discharge relationship, see e.g., Westerberg et al., 2020), whereas the corresponding re-
turn period – and thus the position on the abscissa – is a statistical estimate. If the uncertainty of the return period estimate is
considered (horizontal bars drawn with observations in Figure 13), there is a large overlap between the confidence intervals
of observations and CS also for the Aare River at Stilli. As was the case with rare to very rare floods (previous section), the
slight disagreement between GWEX-based simulations, disaggregation-based simulations, and extrapolations of discharge
observations is not surprising due to the limited length, representativity and homogeneity of the flood records, as well as due
to slightly different reference time periods.

### 5.3 Differences between GWEX-based and SCAMP-based simulations

As a further check of plausibility, the SCAMP hybrid weather generator based on atmospheric and weather analogues was
used as an alternative for the first link of the model chain. SCAMP is structurally independent from GWEX and makes use
of additional variables stemming from an atmospheric reanalysis (see Section 3.2.2). A full set of 30 scenarios with 10 000



**Figure 13.** Exceedance curves for AMFs from 289 000 years of CS based on GWEX weather generator scenarios (red), split up into blocks with length equal to that of observed peak flow records. Results are shown for the Aare River at Aarburg (a, block length 100 years), the outlets of rivers Reuss (b, 106 y) and Limmat (c, 65 y), and for the Aare River close to the outlet at Stilli (d, 112 y). Orange: AMFs from 85 years of simulation (median representative parameter set) using disaggregated weather observations 1930–2014 (DISAG); black: top-5 observed peak flows drawn at return periods estimated by FOEN (Baumgartner et al., 2013); blue: extrapolation of observed peak flow records by FOEN (Baumgartner et al., 2013); green: regionally enhanced extrapolation of observed peak flow records according to EPFL (Asadi et al., 2018); light brown (for Aare at Aarburg only): range of reconstructed historical floods (Baer and Schwab, 2020). Measurement sites do not always match simulation sites exactly; the corresponding observations and extrapolations have been scaled where necessary, assuming constant discharge per unit area.

years of hourly data was produced in SCAMP and evaluated meteorologically (see Sections 4.1 and 5.1). However, it was not possible to run all these data through the hydrological model and routing due to high computational cost. Instead, a number of large precipitation events were sampled from the SCAMP scenarios, looking at the highest cumulative precipitation over 1, 3, 7, 30 and 60 days in the entire Aare River basin as well in five large sub-regions. For each accumulation period and perimeter, the 300 largest events were identified, leading to a total of 3 425 events after elimination of duplicates. These events were then run through the hydrological model and routing. To avoid assumptions about initial conditions, a warm-up period of 10 years was used, consisting of the SCAMP scenario data preceding the respective event. Note that this sampling

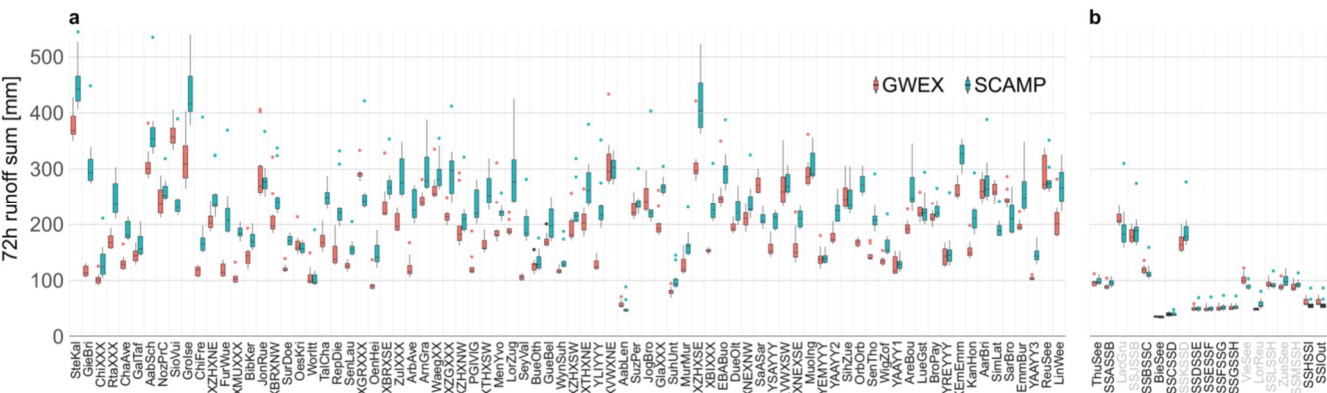

**Figure 14.** Comparison of hydrological results achieved with GWEX inputs (red) and SCAMP inputs (blue). Shown are boxplots for the ten largest 72-hour runoff sums (median representative parameter set), each for the outlets of the 79 individual HBV sub-catchments (a, ordered by increasing catchment area) and the 19 RS Minerve nodes (i.e., critical sites including Aare River outlet) (b, ordered by location along the Aare River, tributaries are designated with grey label colour). For the sub-catchment and output node abbreviations see Supplementary Tables 2 and 3.

did not systematically consider contributions of snowmelt, and that it is therefore not possible to make robust estimates regarding the return periods of the ensuing hydrological events. Furthermore, the largest meteorological events do not necessarily lead to the largest hydrological events. Comparison of the simulations using selected SCAMP scenarios with the full simulations using all GWEX scenarios could thus be misleading. The evaluation was therefore limited to the 10 largest discharge events resulting from SCAMP- and GWEX-based simulations, respectively.

Although SCAMP and GWEX are methodologically fully independent from each other, their largest precipitation events led to simulated floods in a similar order of peak magnitude (Figure 14). In the individual HBV sub-catchments, SCAMP generally led to slightly higher peak discharges than GWEX; this difference was smaller in large sub-catchments. Simulation results after hydrological routing compare equally well, also with a tendency of SCAMP scenarios leading to slightly higher peak flows in comparison to GWEX scenarios. The seasonality of the 10 largest events is limited to May–October using GWEX and to May–November using SCAMP. GWEX shows a slight skew towards June–August and SCAMP towards August–October. These differences are at least partly explained by the constraint that only a subset of SCAMP events could be examined, and that the selection of this subset has some limitations as noted before.

## 5.4 Comparison to PMP-PMF approach

The so-called PMP-PMF method is popular in many countries as a basis for safety assessments of dams and critical infrastructure. In this method, the estimated probable maximum precipitation (PMP) of a certain duration serves to estimate an associated probable maximum flood (PMF) (Kienzler et al., 2015; Felder and Weingartner, 2017). Several methods for making PMP-PMF estimates have been proposed, but there is no scientific consensus about a preferred method. Large uncertainties are inherent in PMP and PMF estimates, e.g. due to the parameterisation of the numerical atmospheric and hydrological models used to simulate them, or due to assumptions regarding the simulated atmospheric configuration and the antecedent





saturation configuration of the catchment. It is thus generally recommended to evaluate their plausibility by comparison to
results from other methods. In recent publications, the PMP-PMF method was applied to Swiss catchments of different sizes
and characteristics, e.g., to the Aare River upstream of Bern (Felder and Weingartner, 2016, 2017; Zischg et al., 2018), the
Kander River upstream of Hondrich (Felder et al., 2019), the Emme River upstream of Wiler (Felder et al., 2019) and the
Sihl River upstream of Zürich (Kienzler et al., 2015). In all of these studies, the PMP estimates have been distributed in time
and space and then run through the hydrological model PREVAH (Viviroli et al., 2009a) to arrive at PMF estimates. Except
for the Sihl River study, a one-dimensional hydrodynamic model was used as a last step to account for effects of overbank
flow and to achieve a more realistic routing.

All these PMF values are based on methods substantially different from the ones employed here, and it is not possible to de-
termine an exact return period for them. However, they are in a similar order of magnitude as the maximum peak flows from
CS based on GWEX scenarios (Table 2). For the Aare River at Bern and the Emme River, values are even very similar,
whereas differences are a little larger for the Kander and Sihl Rivers. The latter two have a comparatively small surface area
and are therefore more sensitive to differences in precipitation configuration present between temporally and spatially redis-
tributed PMP values and weather generator outputs. Moreover, it has to be kept in mind that GWEX was developed with a
focus on larger regions (area of roughly 1 000 km² or more) and thus for combinations of the sub-catchments used in the
present study. Peak flow results from single sub-catchments should only be interpreted with the greatest care. On the other
hand, the PMP maps elaborated for Switzerland cannot be used to derive PMFs for large catchments: In these maps, the PMP
estimates reported for the different locations of a given area often result from different large-scale atmospheric configura-
tions that are highly unlikely to occur at the same time. For example, the 24-hour PMP in the southern part of the Swiss Alps
is a compound of wind flow from SW to SE, whereas in the northern part it is a compound of wind flow from W, NW, NE
and E. In addition, the upper recommended spatial scale for use of the PMP-PMF estimates is a few 100 km² (e.g., 230 km²
for Fallot et al., 2017). Notwithstanding, the rough agreement between the PMP-PMF estimates available and results of the
CS approach strengthens confidence in view of the fundamental methodological differences. Results from using SCAMP

**Table 2.** Probable Maximum Flood (PMF) estimates reported in the literature that apply to the perimeter covered in the present study, and
corresponding maximum peak flows from CS based on GWEX ($Q_{max, GWEX}$).

| River and site name | Area [km²] | PMF [m³ s⁻¹] | $Q_{max, GWEX}$ [m³ s⁻¹] |
|---|---|---|---|
| Aare at Bern | 2 969 | 1 296 [a] | 1 250 [d] |
| Emme at Wiler | 939 | 1 388 [b] | 1 356 [e] |
| Kander at Hondrich | 491 | 830 [b] | 1 050 |
| Sihl at Zürich | 336 | 975 [c] | 772 |

[a] Felder and Weingartner (2016, 2017); Zischg et al. (2018)
[b] Felder et al. (2019)
[c] Kienzler et al. (2015)
[d] Results for the Aare River at Halen, which compares well to Aare at Bern (see Supplementary Table 3)
[e] Results for the Emme River outlet, which compares well to Emme at Wiler (see Supplementary Table 3)





scenarios are not available here because the selection of events focused on peak flow values at the outlet of the Aare River basin; this selection does not cover the largest events in individual smaller sub-catchments.

## 5.5    Uncertainties and limitations

The CS approach as implemented here is subject to several uncertainties and limitations. These mainly stem from structural and parameter uncertainties of the weather generator, the hydrological model and the hydrological routing; the limited length of the observations; measurement errors especially in precipitation and discharge; and uncertain stage-discharge relationships. In addition, the approach assumes that key characteristics of the model chain – such as the spatial dependence structure of large precipitation events – are also valid for extreme events well beyond the observed range.

While selected aspects of these uncertainties have been briefly discussed above and are described in more detail elsewhere (e.g., Sikorska-Senoner and Seibert, 2020; Staudinger and Viviroli, 2020; Westerberg et al., 2020; Andres et al., 2021; Sikorska-Senoner, 2021), a full quantification of uncertainties propagated through the model chain was not feasible due to the enormous computational cost of a comprehensive analysis.

When it comes to large simulated discharge extremes in the present domain and scale, the behaviour of the weather generators GWEX and SCAMP has a major impact on results. As a basis for the parameterisation of the two generators, it was possible to use a high-quality data set of meteorological records with a maximum duration of 85 years since 1930. Both length and spatial coverage of this dataset are exceptional in comparison to other regions and allowed for a very robust estimation of the weather generator parameters. However, for the domain of highest extremes the length of the records is still limited and permitted only a partial evaluation of results. In particular, this limits knowledge of the spatial covariance between local extremes. Within the methods employed here, potentially better model configurations could only be found with considerably longer records, which is unrealistic. However, the comparison of peak flow results based on the two methodologically fully independent weather generators suggests that structural choices are not decisive. In summary, GWEX and SCAMP are two valid weather generators that are used to represent the structural uncertainty in the meteorological part of our study. Both generators have been evaluated in depth.

Regarding hydrological modelling and routing, it should be mentioned that structural uncertainty can surmount parameter uncertainty (Vrugt et al., 2003; Kavetski et al., 2006; Schaefli et al., 2007; Sikorska and Renard, 2017). For reasons of computational cost, it was only possible to quantify parameter uncertainty at individual sub-catchments and provide analyses on stage-discharge uncertainty at selected sites (Westerberg, 2020; Westerberg et al., 2020). However, high runoff coefficients can be expected for rare to very rare flood events like the ones in focus here, and consequently the magnitude of simulated precipitation (including associated uncertainties) is likely more decisive than the hydrological model structure. In addition, we propagated the three representative parameter sets of the hydrological model through to the hydrological routing. These sets are intended to represent the prediction interval due to parameter uncertainty in the hydrological model (Sikorska-Senoner et al. 2020) at individual sub-catchments. However, the cumulative effect of uncertainty propagation through the model



chain at different sites along the major rivers is difficult to assess due to its nonlinearity, particularly because different uncertainty sources may dominate the simulation uncertainty at different sites.

In addition, the peak flow estimates do not distinguish between different flood governing processes such as rainfall-driven or snowmelt-driven floods (Merz and Blöschl, 2003; Diezig and Weingartner, 2007; Sikorska et al., 2015). Flood estimates adapted to specific flood types might improve the realism of the results, but this issue would require further research.

## 6       Conclusions

CS is a valuable option for estimating rare to very rare floods at multiple sites in a large river basin. Compared to statistical
approaches based on streamflow observations, the CS approach has substantial advantages in that it explicitly considers important processes of flood generation such as soil moisture, snow accumulation and snowmelt, and in addition can implement lake regulation, dam operation as well as lake and floodplain retention. Even more importantly, the large diversity of possible but not observed temporal and spatial hydrometeorological configurations (for both antecedent conditions and weather forcing sequences) covered by the simulations provides considerable extra information on the magnitude of floods
with a certain return period. This enables the identification of critical hydrometeorological configurations that could not have been found with a simple a priori guess of a so-called design configuration obtained from relating a design weather event with an assumed initial catchment state.

For return periods larger than roughly 1 000 years, the flood peaks simulated for multiple sites in the Aare River basin are slightly higher than what could be expected from a frequency analysis of discharge observations. This disagreement, how-
ever, is not surprising due to the limited length, representativity and homogeneity of the flood records. A comprehensive assessment of the simulations has not revealed important shortcomings, and plausible explanations were found for the disagreements. Also, the application of two structurally independent weather generators has shown comparable hydrometeorological results, which increases confidence in the flood estimates.

Despite the advantages of the CS-based flood estimation presented here, it should be kept in mind that results are still subject
to considerable uncertainties. These are largely due to the limited length of meteorological and hydrological observations available and can thus not be fully amended with additional computational resources and a higher number of simulated scenarios.

## 7       Outlook

The wealth of hydrometeorological scenarios available from long-term CS at multiple sites in a large river basin opens up
some interesting possibilities. We demonstrated that the present implementation is indeed not only useful for estimation of rare to very rare floods, but also dependable for floods with return periods clearly lower than 1 000 years. There, long-term





CS at multiple sites can be used as an alternative approach to flood estimation and complement the extrapolation of streamflow observations. In particular, CS results are not prone to inhomogeneities due to relocation of streamflow gauges, changes in river network and hydraulic structures. However, CS, of course, contains its own specific set of uncertainties and limita-

tions. Furthermore, flood estimates for the sites considered in CS are inherently consistent because they stem from the same meteorological scenario input. This consistency is important, e.g., for the comparability of hazard mapping over large areas, but is frequently not ensured because of the different record lengths at the relevant observation sites. The same advantages regarding consistency can be exploited for multivariate flood estimates, including flood volumes (e.g., Brunner et al., 2017) and exceedance times of flood levels, and for identifying relevant hydrograph shapes for different return periods, e.g., by

processing the exceedingly long simulations with functional data analysis (Chebana et al., 2012; Ternynck et al., 2016; Staudinger et al., 2021). In this context, an extension of the methods towards small sub-catchments, e.g., 10 km$^2$ or larger, would be highly desirable, since this scale range is even less well covered by streamflow observations. However, this would require considerable methodological adaptations to the weather generators used here. A particular challenge would be to ensure easy application over a wide spatial domain (e.g., all of Switzerland), avoiding time-consuming set-up and calibration

procedures for individual catchments.

A multi-site CS implementation can also inform comprehensive flood risk assessments in a large river basin (see EXAR project, Andres et al., 2021). The abovementioned spatial consistency of results is a decisive advantage for this kind of assessment, and floods even less probable than the ones considered here can be estimated with a focus on critical infrastructure. Since extrapolation of CS results is not advisable due to the large uncertainties involved in the model chain, event tree anal-

yses can be performed based on CS results, and return periods of 100 000–10 000 000 years can be examined (Dang and Whealton, 2020). For this, the hydrometeorological scenarios can be combined with 2D hydraulic simulations covering further relevant hazard scenarios (Pfäffli et al., 2020), including landslides, blockages of bridge openings by driftwood, bank erosion, failure of protective dikes, human failures (e.g., in weir regulation), and dam failures due to extremely rare earthquakes (Andres et al., 2021). Such event tree analyses are not possible on the basis of PMP-PMF or other approaches that do

not include an estimate of the return period.

The weather generator is obviously a key component, enabling the exploration of a large variety of possible hydrometeorological configurations and their development into flood events. However, the choice and implementation of a suitable weather generator is also one of the most challenging issues in the CS approach (Lamb et al., 2016). Such a generator is subject to many requirements that may be difficult to satisfy: It has to produce relevant simulations across the whole continuum

of weather situations from frequent precipitation events to extreme ones, including wet and dry spells; it has to cover extents ranging from localised to catchment-wide; it has to account for the dependence between weather event characteristics and weather types, likely calling for specific parametrisations; and it has – depending on the region – to address the specific challenges of simulating meteorological and hydrological processes in complex terrain. The development of such a generator is thus not straightforward, and more attention should be paid to this issue so that robust and relevant simulation tools can be

made available for similar studies worldwide. The weather generators developed for this work are built on the latest and



most advanced statistical models and developments available to date. If the rare weather scenarios generated for this work are thus very likely relevant from a statistical point of view, their physical relevance remains uncertain. An alternative to weather generators for rare events could be numerical atmospheric models as they allow for a more physically based approach. Numerical atmospheric models are however not free of limitations either, as they are typically based on a number of
simplifications, assumptions and parametrizations. A relevant estimate of critical weather events would also require intensive uncertainty analyses, likely difficult to achieve. More importantly then, the large computational cost of such models prevents the generation of corresponding long time series of weather scenarios. This in turn prevents an in-depth exploration of a large variety of hydrometeorological configurations – combining different spatial patterns of initial hydrological states and different dynamics of weather development, and in turn, the identification of critical hydrological configurations. Besides the
uncertainties of the meteorological scenarios, the uncertainties in the hydrological simulations should also be examined more rigorously in future research. For large-scale applications like the one presented here, computational cost is again a restricting factor to date. The same applies to uncertainty evaluations that propagate from meteorological scenarios to hydrological modelling and routing.

Implementing the present CS for multiple sites in other large river basins is feasible as far as records of discharge, precipita-
tion and temperature are available in sufficient length (ideally a few decades), as well as in appropriately high temporal (hourly) and spatial (see recommendations by World Meteorological Organization, 1994) resolution. However, the specific weather regimes and precipitation characteristics of other regions would likely require an adaptation of the weather generator to well represent precipitation and temperature events that are critical for the region considered.

### Data availability

The resulting data can be obtained from the first author upon reasonable request.

### Author contributions

Conceptualization: ACF, BH, DV, JS, RW; model development, calibration and simulations: AS, DR, JC, GE, GN, MK, MS; data analysis and investigation: AS, CW, DR, DV, GN, GE, JC, MK, MS; visualization: AS, CW, DR, DV, GE, GN, JC, MS, MK; draft writing: DV; review and editing: ACF, AS, BH, CW, GE, GN, JS, MK, MS.

### Competing interests

We declare no competing interests.





**Acknowledgements**

This research has been conducted as a part of the EXAR project (hazard information for extreme flood events on the rivers Aare and Rhine). EXAR was funded by the Swiss Federal Office for the Environment (FOEN), the Swiss Federal Nuclear
Safety Inspectorate (ENSI), the Swiss Federal Office of Energy (SFOE), the Swiss Federal Office for Civil Protection (FOCP) and the Swiss Federal Office of Meteorology and Climatology (MeteoSwiss). We thank MeteoSwiss, FOEN as well as the cantons of Aargau, Bern, Fribourg, St. Gallen, Vaud and Zurich for providing data to make this study possible. We also thank Peiman Asadi, Anthony Davison, Sebastian Engelke, Luise Keller, Thomas Lugrin and Marc Vis for their contributions. The ScienceCloud computational and storage infrastructure provided by Service and Support for Science IT (S3IT)
at the University of Zurich enabled part of the computation-intensive simulations.

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
