# Peer review of "Comprehensive space-time hydrometeorological simulations for estimating very rare floods at multiple sites in a large river basin"

_Natural Hazards and Earth System Sciences, 2022_

## Author Comment (AC1)

**Response to anonymous referee #1**

We thank referee #1 for reading the manuscript carefully and providing thoughtful and constructive comments. Below, their comments (C) are reproduced with our responses (R) following. Manuscript text is shown in serif font, intended changes and additions are underscored.

C1.1 Line 53: It could be worth mentioning here the work on wave superposition of Guse et al (2020).

R1.1 Thank you, we will add this reference.

C1.2 Line 123: Do these 79 sub-catchments correspond to the 19 sites where the whole simulation chain was evaluated? Please clarify.

R1.2 The 79 sub-catchments do not correspond to the 19 critical sites, but were combined with hydrological routing to provide results at the 19 critical sites. We will clarify this by adding a sentence and providing references to the relevant sections:

For the present study, we subdivided the Aare River basin into 79 meso-scale sub-catchments with a median surface area of 123 km$^2$ (range: 19.1–1 061 km$^2$) (Figure 1). These sub-catchments were the basis for the hydrological modelling (Section 3.3.1) and encompass regimes dominated to a varying degree by glaciers, snow and rain (Weingartner and Aschwanden, 1992). Outputs of these sub-catchments were subsequently combined with hydrological routing to provide results at 19 critical sites (Section 3.3.2).

C1.3 Line 129-130: From Figure 1 it seems that a portion of stations are located outside of the Aare catchment. Please indicate how many stations were located within.

R1.3 We will add this information to the manuscript text:

The meteorological data encompass continuous records of daily precipitation (1930–2014) at 105 sites (of which 78 are located within the Aare River basin), daily temperature (1930–2014) at 26 sites (17 within Aare River basin), hourly precipitation (1990–2014) at 65 sites (24 within Aare River basin) and hourly temperature (1990–2014) at 67 sites (25 within Aare River basin) (Supplementary Table 1).

C1.4 Line 137-138: According to the Line 130 hourly data is available only from 1990. Please clarify.

R1.4 Lines 130ff. describe the meteorological data (hourly available from 1990), lines 136ff. the hydrological data (hourly available from 1974). To clarify, we will slightly adjust the description of hydrological data:

The hydrological data encompass continuous discharge records at 65 stations (Supplementary Table 2). The hourly hydrological data (1974–2014) have a median length of 36 years, with a range of 15–41 years, the daily hydrological data (1930–2014) also have a median length of 36 years, but with a range of 16–85 years.

C1.5 Line 141-144 and later: It is not clear how these hydraulic simulations were used in this study. Please clarify.

R1.5 The hydraulic simulations served to parameterise the hydrological routing with RS Minerve, as explained further below in Section 3.3.2. We will clarify this with the following addition:

These simulations represent the behaviour of the river system at flows with return periods of 100, 1 000 and 10 000 years, particularly as regards bank overflow and floodplain retention (Staudinger and Viviroli, 2020), and were used to parameterise the hydrological routing (Section 3.3.2).

C1.6 Line 145: What about the remaining regulated lakes? Was information about their management not available? How their regulation was accounted in the hydrological simulations? Please clarify.

R1.6 Further regulated lakes are only Lake Ägeri (7.2 km²) and Lake Lauerz (3.1 km²). Implementation of a regulation scheme was not deemed necessary for these lakes because of their comparatively small area, and because their outflow is attenuated by another (much larger) regulated lake further downstream (Lake Zug and Lake Lucerne, respectively).

The remaining lakes with an area of at least 0.1 km² are not regulated (Lake Walensee 24.1 km², Lake Sempach 14.5 km², Lake Baldegg 5.3 km², Lake Hallwil 10.3 km², Lake Sarnen 7.5 km², Lake Oeschinen 1.1 km²). The outflow of the largest of these unregulated lakes, Lake Walensee, is attenuated by the regulated (and considerably larger) Lake Zurich further downstream.

We will clarify the text as follows and add information on the surface area of the lakes:

Regulation rules for the large lakes (Lake Biel, surface area 39.8 km²; Lake Brienz, 29.8 km²; Lake Lucerne, 113.6 km²; Lake Thun, 48.4 km²; Lake Zug 113.6 km²; Lake Zurich, 90.1 km²) were provided by the corresponding authorities. Depending on the lake, the rules are aimed at diverse and partly contradicting targets such as protecting settlements down-stream from floods, avoiding inundation of the lakeside areas, preserving habitats, keeping natural stage fluctuations and ensuring lake navigation. The information available ranged from detailed stage-discharge diagrams at a daily or monthly scale to rough indications of target discharge values for different intervals of lake level. For the two remaining small regulated lakes (Lake Ägeri, 7.2 km²; Lake Lauerz, 3.1 km²), the impact of regulation on flows at the critical sites was considered minor and thus neglected, also because in both of these cases, another regulated lake with considerably larger surface area is located downstream.

C1.7 Table 1 a: Are these actually observations or simulations by a hydraulic model? Please clarify.

R1.7 These are observations that were routed downstream (from the observation sites to the sites where historical reconstructions were feasible) with the hydraulic model BASEMENT. We will slightly extend the two footnotes to clarify:

[a] values for 2005 and 2007 are observations for Aare at Brügg-Aegerten that were routed downstream to Solothurn with BASEMENT (see Baer and Schwab, 2020; Pfäffli et al., 2020)

[b] values for 2005 and 2007 are observations from sites Aare at Murgenthal, Wigger at Zofingen and Dünnern at Olten that were routed downstream to Aare at Olten with BASEMENT (see Baer and Schwab, 2020; Pfäffli et al., 2020)

C1.8 Line 156-159 and later: At this point the rationale for using two different weather generators is not clear. Only in Lines 710-713 the reason for using two of them is stated. Please add a clarification earlier in the manuscript to clarify the use of two methods.

R1.8 For clarification, we will move Lines 710–713 to the section containing Lines 156–159, slightly adjust the text and add cross-references:

Our model chain consists of three main components. First, two weather generators – GWEX (Section 3.2.1) and SCAMP (Section 3.2.2) – were used to provide 30 time series scenarios of precipitation and

temperature with a length of 10 000 years each, and to assess the structural uncertainty in the meteorological part of this study (Sections 5.3 and 5.5). Second, the full outputs of GWEX were used as input for the bucket-type catchment model HBV [...]

C1.9 Line 160: From the description later in the manuscript it seems to me that the complete years with largest events were selected and not events per se. Please clarify this to omit any confusion.

R1.9 That is indeed the case, thank you for pointing this out. We will clarify as follows, also adding that precipitation events were in focus:

From SCAMP, selected scenario years that contain large precipitation events were also run through the HBV model (Section 5.3).

C1.10 Line 163: Please indicate how these 19 sites were selected.

R1.10 That information is already given in Section 3.3.2 on RS Minerve: "The output nodes themselves were set at locations where river valley morphology prevents extensive floodplain inundation, and thus all discharge flows through the main river channel. This procedure was motivated by the need to partition 2D hydraulic modelling in EXAR into independent subsystems (Pfäffli et al., 2020)." We believe that Section 3.3.2 is the appropriate location for this information, and that it would be too detailed for Section 3.1 (that contains Line 163). For clarity, we will add a cross-reference to said Line 163:

The final simulation outputs span roughly 300 000 years at an hourly time-step and cover 19 critical sites (including the Aare River outlet, see Section 3.3.2) as well as the outlets of the 79 sub-catchments simulated with HBV (Figure 2, Supplementary Tables Supplementary Table 2 and Supplementary Table 3).

C1.11 Line 182-183: This is not clear. Please provide more detailed information on hourly disaggregation since hourly simulations were indicated as one the main novel point of this study.

R1.11 We agree that additional information should be provided on this particular aspect, which was not described in the references provided. The following paragraph will be added to the revised manuscript under a new subsection 3.2.3:

3.2.3 Temporal disaggregation

For the hydrological simulations at sub-catchment scale, sub-daily data were needed. For both GWEX and SCAMP, this disaggregation was based on weather analogs: For each day in the simulation period, the daily weather variables obtained with the weather generator were disaggregated according to the spatial and temporal structure of an analogous day for which hourly observations were available for the period 1990–2014. The analog day candidates were identified using a distance criterion (the Root Mean Square Error, RMSE) which measures the similarity between the regional weather situation of the target generated day and that of the observations. For GWEX, the regional weather situation of a given day was described by the spatial field of the weather variable considered, namely the daily value of the variable available at the multiple gauge stations in the area. For SCAMP, the regional weather situation was described with the regional MAP and MAT scenarios generated in the first step of the generation process. This disaggregation approach directly exploits past observed spatial-temporal structures and imposes the relative distribution of the candidate day (i.e., the sub-daily distribution of the precipitation and temperature values, as well as the sub-regional distribution, so-called fragments) on the day of interest. This approach, also called the "method of fragments" (Buishand and Brandsma, 2001), is described in detail in Appendix 10.2 of Staudinger and Viviroli (2020) for GWEX scenarios and in Mezghani and Hingray (2009) for the simulation context of SCAMP.

Several lines in sections 3.2.1 and 3.2.2 were referring to these disaggregation steps and will be removed: "The parameters of GWEX were defined based on daily observed weather 1930–2014, and its outputs were then further disaggregated to hourly values with the help of hourly observations 1990–2014." in Section 3.2.1 and "For the hydrological simulations at sub-catchment scale, sub-daily data were needed. To this end, a non-parametric disaggregation approach was applied, following the methodology developed by Mezghani and Hingray (2009) for the upper Rhone River and using sub-daily observations available for a limited set of stations 1990–2015." in Section 3.2.2.

C1.12 Line 200-202: Please provide more information on how hourly discretization was done. The provided references focus on daily simulations.

R1.12 Please see our response to comment C1.11.

C1.13 Line 215-216, 233-235: It is still not clear to me if the weather generators produce areal precipitation directly or at-site simulations for multiple sites that are later used to produce areal mean. Please clarify.

R1.13 The information about how MAP and MAT values are obtained is indeed missing from the current manuscript. Therefore, a new subsection 3.2.4 will be added:

3.2.4. Mean areal values

As the HBV model requires mean areal values of precipitation and temperature as inputs for a given sub-catchment (i.e., MAP and MAT), we have processed the outputs of the two weather generators as follows: For GWEX, the simulated MAP and MAT values for each sub-catchment are obtained using the Thiessen (1911) polygon method applied to the weather scenarios produced at the multiple sites. For SCAMP, simulated MAP and MAT are directly obtained from the spatial-temporal disaggregation of MAP and MAT values generated at the regional scale (see Section 3.2.3).

C1.14 Line 224-226: Was the same disaggregation technique used for SCAMP as for GMEX? Please clarify.

R1.14 Please see our response to comment C1.11.

C1.15 Line 226: Should it be 2014 instead?

R1.15 It should, thank you for catching this. This will be amended, also for two more occurrences (Line 340 and caption of Figure 6).

C1.16 Line 241: Why not all of the available 65 gauged catchments were used for calibration? Please clarify.

R1.16 We did not calibrate HBV for sub-catchments with streamflow records that are strongly influenced 1) by lake regulation (sites Aare at Brugg, Aare at Brügg-Ägerten, Aare at Hagneck, Aare at Murgenthal, Aare at Thun, Aare at Untersiggenthal-Stilli, Limmat at Baden, Limmat at Zürich-Unterhard, Lorze at Frauenthal, Reuss at Luzern, Reuss at Mellingen, Saane at Laupen, Saane at Fribourg); and 2) by bank overflow and floodplain retention (as far as not already contained in aforementioned sites: Emme at Wiler). In our study, these effects of lake regulation, bank overflow and floodplain retention were simulated by RS Minerve. The streamflow records not used for calibration of HBV were used for verifying results of RS Minerve at 15 out of the 19 critical sites (see records denoted with asterisk in Supplementary Table 2).

While processing this comment, we have noted that Supplementary Table 2 needs amendment for two entries: "Aare at Bern, Schönau" should have an asterisk (only used for validation of RS Minerve), and Galtera at Tafers should not be listed (only comparatively short streamflow record that was not used for calibration of HBV). As a consequence, the median catchment area of HBV calibrated catchments needs to be corrected from 115 to 117 km².

We will fix Supplementary Table 2 and clarify the text on Line 241 as follows (implementing the amendment to median catchment area):

A total of 49 gauged sub-catchments (median catchment area: 117 km²; see Figure 1) were calibrated, focusing on sites without major impact of lake regulation, bank overflow and floodplain retention.

C1.17 Line 255-259: In my opinion this description is not very helpful for understanding the functionality of the routing procedure and how it was parametrized. Consider providing a more detailed description.

R1.17 We will rephrase and expand the description as follows:

Discharge of the individual sub-catchments simulated with HBV light was finally combined and routed using the hydrological routing system RS Minerve (García Hernández et al., 2016). As with HBV, the main reasons for using RS Minerve were its speed and well-documented applications in Switzerland (Horton et al., 2021). The simplified representation of the Aare River system built in RS Minerve emulates more detailed 2D hydraulic simulations of synthetic hydrographs with BASEMENT (see Section 2). To cover a broad spectrum of possible event magnitudes and explore the effect of lake regulations, synthetic hydrographs with return periods of 100, 1 000, and 10 000 years were considered. Major effects of bank overflow and floodplain retention – resulting in attenuation and retardation of the flood peak – were considered across a wide range of discharges by implementing channels both in series and in parallel at relevant sites. These channels account for estimated channel flow capacity and inundated area. Levee breaks, by contrast, have not been implemented.

C1.18 Line 261-262: This is rather vague. Please clarify.

R1.18 We will change the text as follows to clarify:

For all of the nine major lakes (Lakes Biel, Brienz, Gruyère, Lucerne, Murten, Neuchâtel, Thun, Zug and Zurich), stage-area-volume relationships were extracted from digital terrain information (swisstopo, 2005). Six of these lakes (Lakes Biel, Brienz, Lucerne, Thun, Zug and Zurich) are regulated, and the regulation rules are usually expressed as stage-discharge relationships, with seasonal, monthly or even daily variation. These rules were digitized and implemented into RS Minerve, where necessary in slightly simplified form. Where available and feasible, rules applied in case of flood events (i.e., deviating from business-as-usual operation) were implemented. For example, discharge in the Aare River is limited to 450 $m^3$ $s^{-1}$ downstream of Lake Thun in Bern and to 850 $m^3$ $s^{-1}$ downstream of Lake Biel in Murgenthal for as long as possible. If the level of a lake rises above the flood stage (i.e., the level above which widespread inundations occur, see flood danger levels defined by Federal Office for the Environment, 2022), stage-discharge relationships as simulated in BASEMENT assuming open weirs are used.

C1.19 Line 268: Please elaborate on the technical issues that occurred.

R1.19 The most likely cause of the issue was a file transfer problem, i.e., the silent crash of a copy process of GWEX data to the computational cloud where HBV was run. Due to this, the HBV simulations for the 11 000 scenario years in question were forced with data from an outdated version of GWEX. The ensuing inconsistency was only discovered a few months later,

when further computationally intensive work had been done within the EXAR project on the basis of the continuous simulations. It was therefore decided to discard the affected scenario years, as the remaining consistent simulations were still 289 000 years long and thus sufficient for the scope of EXAR. We will complement the text as follows:

From the 300 000 years simulated in total, 11 000 were discarded due to an inconsistency (most likely caused by a file transfer problem and the subsequent usage of an outdated version of GWEX data; for details, see Viviroli and Whealton, 2020), leaving 289 000 years for detailed analysis.

C1.20 Line 269: Please indicate how these precipitation events were identified. Was the precipitation depth or volume considered? What was the threshold? Moreover, please also indicate how the initial conditions for these years were handled.

R1.20 The selection procedure for precipitation events is already described on Lines 643–646, as are initial conditions on Lines 646–647. For better readability we will move these descriptions to paragraph 3.3.2 (and thus the section and line the referee refers to) and combine them with the existing text in adjusted form:

From SCAMP, only a selection of scenario years containing the highest cumulative precipitation events were considered and run through the hydrological model and routing due to computational time limitations. For each of five accumulation periods (1, 3, 7, 30 and 60 days) and six perimeters (entire Aare River basin as well in five large sub-regions), the years containing the 300 largest events were identified, leading to a total of 3 425 individual years after elimination of duplicates. To avoid assumptions about initial conditions, a warm-up period of 10 years was implemented using the SCAMP scenario data preceding the respective event.

The text on Lines 643–647 will be shortened to:

Since it was not possible to process all of these data with the hydrological model and routing due to high computational cost, 3 425 years containing the largest precipitation events were sampled from the SCAMP scenarios (see Section 3.3.2) and then run through HBV and RS Minerve. Note that the event sampling did not systematically consider contributions of snowmelt [...]

C1.21 Section 4.1: This section only provides results for daily simulations. Since the focus of this study is actually on hourly simulations the corresponding results at this resolution for weather generators should be provided.

R1.21 Thank you for this comment. The motivation for the evaluation using daily observed values was the length of available observation time series at this time resolution. At a daily scale, most of the stations provide complete observation time series for the period 1930–2014 while hourly observation time series are only available since 1981 and for 60 stations only. The full ensemble of hourly precipitation data used in this study (65 stations) covers the period 1990–2014 which limits their use for the evaluation of precipitation extremes (Figs. 3 and 4). Furthermore, for GWEX, the generation is performed at the daily stations first. When performed at the daily time step, the evaluation of the simulated values against the observed data at these stations (Fig. 5) can thus be directly interpreted as the behavior of GWEX and not the disaggregation method.

We will add the following paragraph at the beginning of Section 4.1 to clarify this point:

This subsection presents an evaluation of the performances of GWEX and SCAMP at the daily scale, by comparison to observations covering the period 1930–2014. Similar evaluations at the hourly scale

are not provided because the shorter period 1990–2014 covered by the hourly observations limits the evaluation of extreme values for large return periods.

C1.22 Figure 4: The resulting spatial pattern of annual maxima from two weather generators appear to be quite different, but very little discussion on that is provided in the text. Consider adding it and indicate what can be the reason for the discrepancies.

R1.22 We agree that these discrepancies should be discussed in the manuscript. We will extend and rephrase the last paragraph of Section 4.1.2 as follows:

At the scale of the entire Aare River basin, MAP extremes are roughly similar for GWEX and SCAMP (Figure 3, Figure 4). At the sub-basin scale, however, the extremes of SCAMP are generally larger than those of GWEX and show slightly different spatial patterns. Both of these differences are probably explained by the fact that the two weather generators are built upon substantially different approaches and generation processes: GWEX produces multi-site 3-day amounts disaggregated at a daily scale, whereas SCAMP produces regional MAP and MAT values at a daily scale. Three-day maxima in SCAMP are thus the result of the aggregation of three consecutive daily simulated values. The temporal coherency between MAP values generated by SCAMP for consecutive days comes from the large-scale atmospheric forcing, which follows relevant atmospheric trajectories from one day to the next. However, this conditioning does not necessarily preserve the day-to-day dynamics of rainfall systems. Nevertheless, it can be noticed that the largest difference – found for the Neuchâtel sub-region – is rather moderate (+10% for MAP3d and +20% for MAP1d). A further comprehensive evaluation of precipitation time series generated with both weather generators is found in Evin et al. (2018, 2019) and Chardon et al. (2020), as well as in Raynaud et al. (2020), which reports on severity, spatial and temporal dynamics, and meteorological relevance of events.

C1.23 Section 4.4: No results for the simulations of the entire chain using SCAMP weather generator are provided. Please add.

R1.23 As stated in Sections 3.1 and 3.3.2, it was necessary to limit SCAMP-based simulations to a selection of years due to computational cost (see also our responses R1.8 and R1.20). There are thus no full continuous simulations of 289 000 years using SCAMP inputs, and the sample of 3 425 SCAMP years run through the entire chain is subject to some limitations, as explained on Lines 647–650: "Note that this sampling [of SCAMP scenario data] did not systematically consider contributions of snowmelt, and that it is therefore not possible to make robust estimates regarding the return periods of the ensuing hydrological events. Furthermore, the largest meteorological events do not necessarily lead to the largest hydrological events." On Lines 650–651 it is thus concluded: "Comparison of the simulations using selected SCAMP scenarios with the full simulations using all GWEX scenarios could thus be misleading. The evaluation was therefore limited to the 10 largest discharge events resulting from SCAMP- and GWEX-based simulations, respectively." For this reason, results of the entire modelling chain are only shown for GWEX in Section 4.4. The 10 largest events from the entire modelling chain using SCAMP as a first chain link are then shown in the sense of a discussion vis-à-vis GWEX-based runs in Section 5.3.

We will add the following text to the beginning of Section 4.4 for clarification:

When running the full hydrometeorological model chain with weather generator scenarios instead of observed weather, there are obviously no reference observations available for evaluating streamflow results. The focus was therefore put on two selected aspects of streamflow and flood behaviour, namely cumulative frequency of streamflow and seasonality of AMFs. The following evaluations are based on the full 289 000 years of CS using GWEX inputs. Selected results on the basis of SCAMP inputs are

only discussed in Section 5.3 since it was necessary to limit simulations to a sample of 3 425 years containing the largest cumulative precipitation events (see Sections 3.1 and 3.3.2).

C1.24 Section 4.4: This result section provides quite a few methodological details. Consider moving them to a corresponding section in the method.

R1.24 Thank you for noting this. We have checked what details seem excessive for a discussion of results and will move the characterisation of the FOEN and EPFL extrapolation methods to Section 2 (see R1.25 below). For the remaining details (such as on seasonality, flow duration curves, annual maximum floods and confidence intervals), we suggest keeping them in Section 4.4 because they are specific to the comparisons done. It might be less convenient to collect the dispersed details from the Methods Section, also because these details are small and do not tie very well to other items presented in the methods. The same applies to the (indeed quite long) legend of Figure 7 where we believe it would be less convincing if details would have to be sought elsewhere (e.g., how the confidence intervals shown for GWEX-based runs were determined: this is a single reference to Loucks and van Beek, 2017 that is not used elsewhere).

The move of event selection details from Section 5.3 to Section 3.3.2 (see R1.20) follows the rationale of this comment.

C1.25 Line 417-419: Please clarify what are these extrapolations.

R1.25 We will clarify which extrapolations are in focus by rephrasing as follows:

Downstream at Golaten, CS is generally higher than observations, FOEN extrapolations (Baumgartner et al., 2013) and EPFL extrapolations (Asadi et al., 2018), while EPFL extrapolations are clearly lower than CS, observations and FOEN extrapolations for return periods larger than 100 years.

At the same time, we will move the characterisation of the EPFL extrapolation ("regionally enhanced") to Section 2 and slightly expand it, and add a characterisation of the FOEN extrapolation. The text on Line 136ff. will be completed as follows:

The hydrological data encompass continuous discharge records at 65 stations (Supplementary Table 2). The hourly hydrological data (1974–2014) have a median length of 36 years, with a range of 15–41 years, the daily hydrological data (1930–2014) also have a median length of 36 years, but with a range of 16–85 years. In addition, records of annual maximum floods were available for some of these stations. These records date back even further, with a median length of 94 years and a range of 32–111 years. For the 44 streamflow measurement stations operated by the Federal Office for the Environment (FOEN), two different extrapolations of the observed annual maximum floods were available: The FOEN approach (Baumgartner et al., 2013), which extrapolates at-site measurements via the Generalized Extreme Value (GEV) distribution, and the EPFL approach (Asadi et al., 2018), which combines at-site measurements with measurements from a group of similar catchments for fitting the parameters of the GEV distribution.

C1.26 Section 5.5: I miss any mentioning about the uncertainties arising from disaggregation from daily to hourly values. Please add.

R1.26 We agree that some limitations of the disaggregation should be acknowledged. A first obvious limitation is related to the observations available at the hourly scale. As a result, for instance, disaggregated fields might miss the spatial-temporal dynamics of localized precipitation events (e.g., convective storms). For temperature, the inclusion of additional predictors

such as the daily temperature field or a preselection of the analog dates based on a seasonal filter or an atmospheric circulation model could also be considered as possible refinements.

Nevertheless, also note that the disaggregation is stochastic. For both GWEX and SCAMP, the spatial-temporal field used for the disaggregation is that of an analog day, which was randomly drawn from a set of analog candidates. This partly accounts for uncertainties in the disaggregation process.

The following paragraph will be added to Section 5.5. after Line 713:

Concerning the temporal disaggregation of the weather scenarios from a daily to an hourly scale, an obvious limitation is related to the limited observations available at the hourly scale. As a result, for instance, disaggregated fields might miss the spatial-temporal dynamics of localized precipitation events (e.g., convective storms). For temperature, the inclusion of additional predictors such as the daily temperature field or a preselection of the analog dates based on a seasonal filter or an atmospheric circulation model could also be considered as possible refinements. Nevertheless, it is also important to note that this disaggregation approach is stochastic, which partly handles the uncertainties related to this postprocessing step. Indeed, for both GWEX and SCAMP, the spatial-temporal field used for the disaggregation is that of an analog day, which was randomly drawn from a set of analog candidates.

C1.27 Line 117: the Rhine River

R1.27 Thank you, will be amended.

C1.28 Line 391: the largest

R1.28 Thank you, will be amended.

**References**

Buishand, T. A. and Brandsma, T.: Multisite simulation of daily precipitation and temperature in the Rhine Basin by nearest-neighbor resampling, Water Resour. Res., 37, 2761–2776, https://doi.org/10.1029/2001WR000291, 2001.

Federal Office for the Environment: The 5 flood danger levels, https://www.hydrodaten.ad-min.ch/en/the-5-flood-danger-levels.html, last access: 22 June 2022.

Mezghani, A. and Hingray, B.: A combined downscaling-disaggregation weather generator for stochastic generation of multisite hourly weather variables over complex terrain: Development and multi-scale validation for the Upper Rhone River basin, J. Hydrol., 377, 245–260, https://doi.org/10.1016/j.jhydrol.2009.08.033, 2009.

Staudinger, M. and Viviroli, D. (Eds.): Extremhochwasser an der Aare: Detailbericht A Projekt EXAR. Hydrometeorologische Grundlagen, Zürich, https://doi.org/10.5167/UZH-201388, 2020.

swisstopo: DHM25: Das digitale Höhenmodell der Schweiz, Wabern, 2005.

Thiessen, A. H.: Precipitation Averages for Large Areas, Monthly Weather Review, 39, 1082–1084, 1911.

---

## Author Comment (AC2)

**Response to anonymous referee #2**

We thank referee #2 for reading the manuscript carefully and providing thoughtful and constructive comments. Below, their comments (C) are reproduced with our responses (R) following. Manuscript text is shown in serif font, intended changes and additions are underscored.

C2.1 Introduction. I have the feeling that only the most standard methods have been referenced in the introduction (conventional flood frequency analysis, regional flood frequency analysis, use of historical information etc.), but in the literature other approaches linking flood estimation with physical processes are present (see e.g., Basso et al. (2021) and references herein for a systematized description of a mechanistic-stochastic physically-based approach for the estimation of river flows/floods). I would suggest the authors to mention other-than-conventional and widely used approaches in the paper introduction, especially if relevant in the discussion of physically-based models or methods less affected by the time series shortness, as the standard ones usually are.

R2.1 Thank you for pointing this out and bringing up alternative approaches.

As for the introduction in general, we indeed tried to mention the most common approaches relevant in our context, namely with focus on applications in large river basins, and on rare to very rare events. We will clarify this and at the same time add some key references concerning flood estimation in a broader context.

As for the mechanistic-stochastic physically-based approach, we will add a paragraph after L55 to mention potential benefits and limitations in the context of our research, following Basso et al. (2021). The preceding text as well as the subsequent introduction to the continuous simulation approach starting on L56 will be slightly reorganised and extended to make it fit to the new paragraph.

Overall, we will modify L40–67 of the introduction as follows:

Generally speaking, common approaches for flood estimation can be categorised into statistical and deterministic (or hydrological) methods as well as combinations thereof (for an overview and evaluation see e.g., Rogger et al., 2012; Okoli et al., 2019). Statistical approaches are widely used (see e.g., Castellarin et al., 2012; Deutsche Vereinigung für Wasserwirtschaft, Abwasser und Abfall, 2012; England, Jr. et al., 2019; Environment Agency, 2020) and also popular to derive design floods for safety assessments. For this, conventional frequency analysis is performed on observed streamflow records, and then a simple return period conversion factor given by design codes (e.g., Bundesministerium für Land- und Forstwirtschaft, Umwelt und Wasserwirtschaft and Technische Universität Wien, 2009; Bundesamt für Energie, 2018; International Commission on Large Dams, 2018) is applied. In addition, it is possible to augment flood frequency analysis with additional data and evidence (Gutknecht et al., 2006; Merz and Blöschl, 2008) such as historical floods (e.g., Bayliss and Reed, 2001; Neppel et al., 2010; Hall et al., 2014; Benito et al., 2015; Salinas et al., 2016; Wetter, 2017), paleofloods (Benito and Thorndycraft, 2005; Baker, 2008; Baker et al., 2010; Benito and O'Connor, 2013; O'Connor et al., 2014), regional frequency analyses (Hosking and Wallis, 1993, 1997), envelope curves (Castellarin et al., 2005), or by differentiating for flood-generating mechanisms (Fischer, 2018; Barth et al., 2019). Also, floods can be estimated from rainfall information via simple approaches such as the GRADEX method (Guillot and Duband, 1969; Naghettini et al., 1996) or the rational method (Mulvany, 1851). Nevertheless, the comparatively short streamflow records contain a rather heterogeneous and likely unrepresentative sample of floods, and neither of the aforementioned methods is able to cover the whole gamut of possible hydrometeorological patterns and the corresponding responses of the river system.

This issue has even greater relevance in large river basins, where flows from individual tributaries interact in a complex manner (see Guse et al., 2020), possibly further complicated through flow management (e.g., lake regulation and reservoir operation).

While the above approaches are predominantly based on statistical elements, further approaches have emerged that combine random elements with understanding of the most relevant physical factors such as soil moisture and runoff dynamics (see e.g., Laio et al., 2001; Porporato et al., 2004; Botter et al., 2007, 2009; Basso et al., 2015, 2016; Zorzetto et al., 2016). Linked with a systematic description of advances in this field, Basso et al. (2021) recently introduced the PHysicallybased Extreme Value (PHEV) distribution as an example of such a mechanistic-stochastic and physically based approach. PHEV showed lower uncertainty and less bias in estimation of large floods (return period of 1 000 years, daily time scale) in comparison to conventional frequency analysis, albeit with a tendency for a slight underestimation and higher variability in performance. Main limitations of PHEV are the assumption of an invariable recession coefficient as well as the exclusion of some hydroclimatological regimes (in particular snow- and glacier-dominated, monsoon and seasonally dry).

Another common approach used in safety assessments are PMP-PMF (Possible Maximum Precipitation-Possible Maximum Flood) estimates, which can follow deterministic (hydrometeorological) or statistical concepts (World Meteorological Organization, 2009). This approach can achieve the range of peak flow extremes examined here, but results have no clear estimate of return period and are usually not applicable over large spatial domains. Moreover, the estimation of PMP and ensuing PMF bears substantial simplifications and considerable uncertainties (Salas et al., 2014; Micovic et al., 2015; Ben Alaya et al., 2018; Zhang and Singh, 2021)

Hydrological methods avoid the abovementioned limitations, more comprehensively link flood estimation with physical processes, and allow for representing effects caused by operation of hydraulic infrastructure. Such methods typically involve a catchment runoff model that is fed with meteorological data and provides simulated discharge as an output (Beven, 2011). In case continuous simulation (CS) is employed rather than an event-based approach, there is no need to separate discharge into baseflow and stormflow, and assumptions about antecedent conditions of a flood event (e.g., snowpack, soil moisture, storage levels of lakes and reservoirs) are not required (Calver and Lamb, 1995; Pathiraja et al., 2012). Beven (1987) was one of the first to recognise the potential of this compelling approach, and CS has indeed been implemented in numerous studies since. However, application in industry is still challenging due to the considerable effort necessary (see overview by Lamb et al., 2016 and references therein). In CS, precipitation data are required to perform rainfall-runoff simulations and subsequently process the simulation results with conventional frequency analyses. Although observed series of precipitation can be used [...]

C2.2 Study area and observational data. Why do time series end in 2014? Are there no more recent data available?

    R2.2. The research presented in this paper was done within the EXAR project, for which work started in early 2016. At that time, consolidated data were available until the end of 2014 only.

C2.3 Methods. What is exactly the rationale behind the choice of using two different weather generators? I understand that they are implemented independently, and they are used for different purposes, but I miss a clear explanation of the reasons why for example you choose GWEX instead of the SCAMP as input to the HBV model and not the other way around.

    R2.3 The two different weather generators are first of all used to assess the structural uncertainty in the meteorological part of this study. They are actually used for similar purposes but, as indicated on L266–270 in the original manuscript, it was only possible to run the full set of

GWEX generated weather scenarios through HBV light and RS Minerve due to the exceptionally high computational cost of long simulations at multiple sites.

Please see also in particular our response to comment C1.8 by referee #1 (who also addressed the rationale for using two different weather generators): For clarification, we will move Lines 710–713 to the section containing Lines 156–159, slightly adjust the text and add cross-references:

Our model chain consists of three main components. First, two weather generators – GWEX (Section 3.2.1) and SCAMP (Section 3.2.2) – were used to provide 30 time series scenarios of precipitation and temperature with a length of 10 000 years each, and to assess the structural uncertainty in the meteorological part of this study (Sections 5.3 and 5.5). Second, the full outputs of GWEX were used as input for the bucket-type catchment model HBV [...]

C2.4 L193: It is not clear to me what do you mean by "…represents the dependence structure of innovations in the generation process"

R2.4 "Generation process" refers to the multivariate autoregressive model (MAR) process which contains so-called innovations which are modelled using a multivariate distribution. "Generation" will be replaced by "multivariate autoregressive model (MAR)":

A Student copula represents the dependence structure of innovations in the multivariate autoregressive model (MAR) and introduces a tail dependence between at-site extremes.

C2.5 L268: could you be clearer about the "technical issues"? What are they related to?

R2.5 This point was also raised by referee #1, see our response to comment C1.19: The most likely cause of the issue was a file transfer problem, i.e., the silent crash of a copy process of GWEX data to the computational cloud where HBV was run. Due to this, the HBV simulations for the 11 000 scenario years in question were forced with data from an outdated version of GWEX. The ensuing inconsistency was only discovered a few months later, when further computationally intensive work had been done within the EXAR project on the basis of the continuous simulations. It was therefore decided to discard the affected scenario years, as the remaining consistent simulations were still 289 000 years long and thus sufficient for the scope of EXAR. We will complement the text as follows:

From the 300 000 years simulated in total, 11 000 were discarded due to an inconsistency (most likely caused by a file transfer problem and the subsequent usage of an outdated version of GWEX data; for details, see Viviroli and Whealton, 2020), leaving 289 000 years for detailed analysis.

C2.6 Results. There is not a clear discussion about the reasons why the two weather generators provide different precipitation ranges. I think that you should spend some time on better describing the differences in the outputs obtained through the simulations and what they are related to.

R2.6 This is a relevant comment which was also raised by referee #1, please see our response to comment C1.22: We will extend and rephrase the last paragraph of Section 4.1.2 as follows:

At the scale of the entire Aare River basin, MAP extremes are roughly similar for GWEX and SCAMP (Figure 3, Figure 4). At the sub-basin scale, however, the extremes of SCAMP are generally larger than those of GWEX and show slightly different spatial patterns. Both of these differences are probably explained by the fact that the two weather generators are built upon substantially different approaches and generation processes: GWEX produces multi-site 3-day amounts disaggregated at a daily scale,

whereas SCAMP produces regional MAP and MAT values at a daily scale. Three-day maxima in SCAMP are thus the result of the aggregation of three consecutive daily simulated values. The temporal coherency between MAP values generated by SCAMP for consecutive days comes from the large-scale atmospheric forcing, which follows relevant atmospheric trajectories from one day to the next. However, this conditioning does not necessarily preserve the day-to-day dynamics of rainfall systems. Nevertheless, it can be noticed that the largest difference – found for the Neuchâtel sub-region – is rather moderate (+10% for MAP3d and +20% for MAP1d). A further comprehensive evaluation of precipitation time series generated with both weather generators is found in Evin et al. (2018, 2019) and Chardon et al. (2020), as well as in Raynaud et al. (2020), which reports on severity, spatial and temporal dynamics, and meteorological relevance of events.

C2.7 L370: I would avoid reporting only the Nash-Sutcliffe efficiency values, but at least complement them with another evaluation criterion, as the NSE is not the optimal one when model accuracy needs to be assessed.

> R2.7 That is of course a valid point, thank you. We will report on all three efficiency criteria shown in Figure 6, but for brevity declare median efficiencies. The range can be inferred from Figure 6. We will in addition report these median efficiencies for the three sites in the Emme, Lorze, and Saane Rivers that showed poorer performance:
>
> Results (Figure 6b) show good to very good agreement between observations and simulations (median efficiencies over all three representative parameter sets: NSE 0.83, KGE 0.85, KGE_NP 0.89) for all sites in the Aare, Reuss and Limmat Rivers. The three sites in the Emme, Lorze, and Saane Rivers showed poorer performance (NSE 0.34, KGE 0.65, KGE_NP 0.66).
>
> The abbreviations of the efficiencies used above are currently only declared and used in the legend for Figure 6. Therefore, we will define them also in the main text on L437ff.:
>
> Hydrological simulations for the individual HBV sub-catchments were evaluated based on three criteria: the Nash-Sutcliffe (NSE) (Nash and Sutcliffe, 1970), the Kling-Gupta (KGE) (Gupta et al., 2009) and the non-parametric Kling-Gupta (KGE_NP) (Pool et al., 2018) efficiencies.

C2.8 L418: I suggest the authors to define the FOEN acronym, as it is not clear what you are referring to (I had to go to the Acknowledgments section to understand its meaning).

> R2.8 Thank you for noting this. In response to comment C1.25 by referee #1, we will expand the description of extrapolation methods in Chapter 2 on study area and observational data, and define the FOEN acronym there already.

C2.9 Figure 10a. Despite considering this representation very nice, I have to say that most of the information in the smallest circles is lost. I would suggest leaving it like it is for the entire basin and the sub-regions but simplify the symbols for all the other sites (maybe only showing a couple of representative durations, so that the colors are clear).

> R2.9. We understand this comment and recognize that it is difficult to appraise the information in the smallest circles precisely. However, we argue that the regions with the largest return levels still stand out thanks to all the small circles. This information cannot be inferred in similar detail at the level of sub-regions only. We have tried a number of alternative design solutions, but the current representation still appeared to be best for highlighting the multiscale structure of precipitation extremes, as well as the regions and areas with the highest precipitation amounts.

C2.10 L470: I believe the first comma should be removed

 R2.10 Thank you, will be done.

C2.11 L666: a comma is missed after e.g.

 R2.11 Thank you, will be added

**References**

Barth, N. A., Villarini, G., and White, K.: Accounting for Mixed Populations in Flood Frequency Analysis: Bulletin 17C Perspective, J. Hydrol. Eng., 24, https://doi.org/10.1061/(ASCE)HE.1943-5584.0001762, 2019.

Basso, S., Schirmer, M., and Botter, G.: On the emergence of heavy-tailed streamflow distributions, Adv. Water Resour., 82, 98–105, https://doi.org/10.1016/j.advwatres.2015.04.013, 2015.

Basso, S., Schirmer, M., and Botter, G.: A physically based analytical model of flood frequency curves, Geophysical re-search letters, 43, 9070–9076, https://doi.org/10.1002/2016GL069915, 2016.

Basso, S., Botter, G., Merz, R. and Miniussi, A. (2021). PHEV! The PHysically-based Extreme Value distribution of river flows. Environ. Res. Lett., 16 (12). doi:10.1088/1748-9326/ac3d59

Beven, K. J.: Rainfall-Runoff Modelling: The Primer, 2nd edition, Wiley, Chichester, 2011.

Botter, G., Porporato, A., Rodriguez-Iturbe, I., and Rinaldo, A.: Basin-scale soil moisture dynamics and the probabilistic characterization of carrier hydrologic flows: Slow, leaching-prone components of the hydrologic response, Water Re-sour. Res., 43, 181, https://doi.org/10.1029/2006WR005043, 2007.

Botter, G., Porporato, A., Rodriguez-Iturbe, I., and Rinaldo, A.: Nonlinear storage-discharge relations and catchment stream-flow regimes, Water Resour. Res., 45, https://doi.org/10.1029/2008WR007658, 2009.

Calver, A. and Lamb, R.: Flood frequency estimation using continuous rainfall-runoff modelling, Physics and Chemistry of the Earth, 20, 479–483, https://doi.org/10.1016/S0079-1946(96)00010-9, 1995.

Castellarin, A., Kohnová, S., Gaál, L., Fleig, A., Salinas, J. L., Toumazis, A., Kjeldsen, T. R., and Macdonald, N.: Review of Applied European Flood Frequency Analysis Methods, COST Action ES0901, WG2, Wallingford, Oxfordshire, UK, 130 pp., 2012.

Deutsche Vereinigung für Wasserwirtschaft, Abwasser und Abfall: Ermittlung von Hochwasserwahrscheinlichkeiten: DWA-Regelwerk, Merkblatt DWA-M, 552, Hennef, 90 pp., 2012.

England, J. F., Jr., Cohn, T. A., Faber, B. A., Stedinger, J. R., Thomas, W. O., Jr., Veilleux, A. G., Kiang, J. E., and Mason, R. R., Jr.: Guidelines for Determining Flood Flow Frequency: Bulletin 17C, Version 1.1, May 2019, U. S. Geological Survey Techniques and Methods, Book 4, Chapter 5b, 168 pp., 2019.

Environment Agency: Flood Estimation Guidelines, Technical guidance, 197 08, 129 pp., 2020.

Fischer, S.: A seasonal mixed-POT model to estimate high flood quantiles from different event types and seasons, Journal of Applied Statistics, 1–17, https://doi.org/10.1080/02664763.2018.1441385, 2018.

Laio, F., Porporato, A., Ridolfi, L., and Rodriguez-Iturbe, I.: Plants in water-controlled ecosystems: active role in hydrologic processes and response to water stress, Adv. Water Resour., 24, 707–723, https://doi.org/10.1016/S0309-1708(01)00005-7, 2001.

Okoli, K., Mazzoleni, M., Breinl, K., and Di Baldassarre, G.: A systematic comparison of statistical and hydrological methods for design flood estimation, Hydrol. Res., 50, 1665–1678, https://doi.org/10.2166/nh.2019.188, 2019.

Pathiraja, S., Westra, S., and Sharma, A.: Why continuous simulation? The role of antecedent moisture in design flood esti-mation, Water Resour. Res., 48, W06534, https://doi.org/10.1029/2011WR010997, 2012.

Porporato, A., Daly, E., and Rodriguez-Iturbe, I.: Soil water balance and ecosystem response to climate change, The Ameri-can naturalist, 164, 625–632, 2004.

Rogger, M., Kohl, B., Pirkl, H., Viglione, A., Komma, J., Kirnbauer, R., Merz, R., and Blöschl, G.: Run-off models and flood frequency statistics for design flood estimation in Austria – Do they tell a consistent story?, J. Hydrol., 456-457, 30–43, https://doi.org/10.1016/j.jhydrol.2012.05.068, 2012.

World Meteorological Organization: Manual on Estimation of Probable Maximum Precipitation (PMP), WMO Publ., 1045, 291 pp., 2009.

Zorzetto, E., Botter, G., and Marani, M.: On the emergence of rainfall extremes from ordinary events, Geophysical research letters, 43, 8076–8082, https://doi.org/10.1002/2016GL069445, 2016.